# Dissecting Causal Mechanism Shifts via FANS: Function And Noise Separation

Gyeongdeok Seo [1]   Jaeyoon Shim [1]   Mingyu Kim [1]   Hoyoon Byun [1]   Yonghan Jung [2 †]   Kyungwoo Song [1 †]

## Abstract

Identifying the drivers of causal mechanism shifts, distinguishing functional changes from noise alterations, termed *dissection*, is a critical yet underexplored problem in data science (e.g., biomedical science and manufacturing). This paper introduces a more general and unified framework, the *function and noise separation* (FANS) framework, that detects and dissects shifts in non-additive, non-linear Structural Causal Models (SCMs) beyond existing additive noise models. Our approach is grounded in a theoretical independence criterion, where function shifts induce a statistical dependence between a node's parents and residual noise. Building on this foundation, we develop a practical two-stage algorithm to efficiently detect and dissect these shifts without retraining. Furthermore, we address the complex challenge of simultaneous function and noise shifts, introducing a formal assumption to resolve their inherent non-identifiability. Our results are corroborated by simulations. Our code is available at https://github.com/MLAI-Yonsei/FANS/.

## 1. Introduction

Observed data frequently originates from diverse environments (Mooij et al., 2020; Karlsson & Krijthe, 2023), leading to conditional distribution changes known as causal mechanism shifts (CMS) (Chen et al., 2023). A causal mechanism is the conditional probability of a node given its parent nodes (Mameche et al., 2024; Perry et al., 2022) in a directed acyclic graph (DAG) describing the data generating processes. A CMS is an alteration in this conditional probability. This phenomenon is examined in multiple disciplines, such as *structural breaks* in econometrics (Bai, 1996; Perron

†Co-corresponding authors. [1]Department of Statistics and Data Science, Yonsei University [2]School of Information Science, UIUC, University of Illinois Urbana-Champaign. Correspondence to: Kyungwoo Song <kyungwoo.song@yonsei.ac.kr>, Yonghan Jung <yonghan@illinois.edu>.

*Proceedings of the 43rd International Conference on Machine Learning*, Seoul, South Korea. PMLR 306, 2026. Copyright 2026 by the author(s).

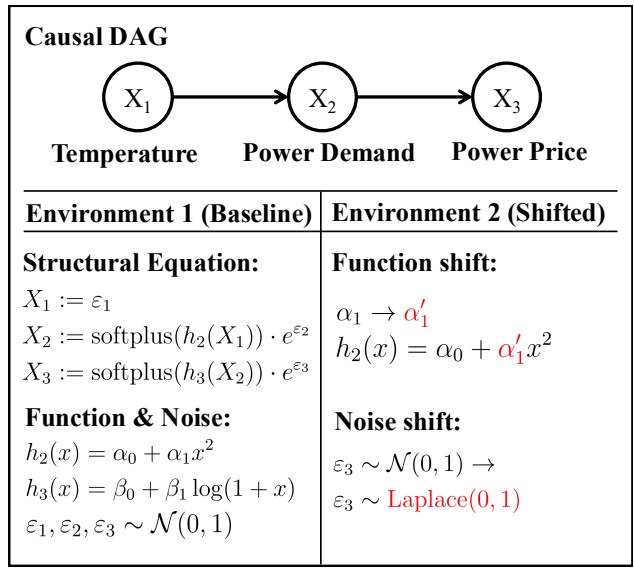

*Figure 1.* Causal graphs depicting how causal relationships vary under different shifts.

et al., 2020) and *conditional distribution tests* in statistics (Hu & Lei, 2024).

Detecting a CMS amounts to pinpointing the node exhibiting a distributional shift in its mechanism across environments. While the detection of CMS and its applications (Reddy & N Balasubramanian, 2024; Mameche et al., 2024; Perry et al., 2022) have been well-addressed (Chen et al., 2023; Mameche et al., 2022), their drivers have received less focus. A CMS can arise from two sources: a shift in the deterministic *causal function* relating variables, or a shift in the distribution of the *exogenous noise* component. A function shift implies that the fundamental relationship between variables has changed. In contrast, a noise shift suggests a change in the influence of unobserved latent variables (Pearl, 2009), warranting investigation into how the underlying context has varied.

Figure 1 illustrates a simple three-node structural equation model for an electricity market with temperature ($X_1$), power demand ($X_2$), and power price ($X_3$). In Environment 1, all nodes follow the same nonlinear, non-additive form. Environment 2 differs from Environment 1 in two ways. First, the causal function from temperature to de-

mand changes: the coefficient $\alpha_1$ in $h_2$ is replaced by $\alpha'_1$, so that the temperature–demand relationship itself (the increase in demand per degree of temperature) is modified. We call this a function shift. Second, the noise distribution of the price node changes from Gaussian to Laplace, while the function remains fixed. This represents a noise shift: the demand–price curve is unchanged, but new, demand-independent shocks—for example, forecast errors from newly introduced renewable energy sources—make price residuals more heavy-tailed.

Distinguishing these sources is crucial, as misinterpretation can lead to flawed conclusions. Consider the simple electricity market model in Figure 1. Suppose that the causal function ($h$) remains unchanged across two countries or time periods, but a new energy source (e.g., large-scale wind and solar generation) is introduced. Its unpredictable forecast errors generate rare but large price spikes, so that the noise distribution changes from approximately Gaussian to a heavier–tailed law such as Laplace. If one ignores this noise shift and keeps assuming Gaussian errors, standard estimation procedures may contort $h$ to absorb these spikes and falsely suggest that the demand–price sensitivity itself has changed. In other words, a change in volatility driven by new energy sources may be incorrectly interpreted as a change in the causal function, leading to misleading policy or market conclusions.

In this paper, we propose a novel framework for dissecting causal mechanism shifts. Specifically, our main contributions are threefold.

First, we relax the restrictive additive noise assumption to accommodate a broader class of causal mechanism. Our method employs Normalizing Flows (NF) to infer a noise component analogous to a residual, applicable to non-additive, non-linear models.

Second, we propose a two-stage framework for detecting and dissecting shifts. The framework trains once on the base environment; new environments require only inference and an independence test.

Third, we address the non-identifiability of simultaneous shifts, where both function and noise change concurrently. We establish formal structural assumptions that make it theoretically possible to distinguish and dissect drivers.

### 1.1. Related Work

Our work addresses the challenge of comparing observational data from two distinct environments, assuming the data are generated by SCMs that differ in their functional relationships or noise distribution. This problem intersects with several lines of research.

**Dissecting Causal Mechanism Shifts.** Dissecting a CMS

involves attributing a change in a conditional distribution to a function shift or a noise shift. This task is simplified under the additive noise model assumption: changes in the conditional mean are attributed to the function, while changes in variance or higher-order moments are attributed to the noise. LinearCCP (Huang et al., 2024) assumes a linear, additive structure, enabling the use of beta-coefficient tests for function shifts and two-sample tests on residuals for noise shifts. Extending to non-linear cases, iSCAN (Chen et al., 2023) can identify functionally shifted edges but still relies on the ANM assumption and an additional inter-parent additivity constraint. The kernel-based approach in (Li et al., 2023) tests for functional changes within the ANM framework but may impose further constraints like Gaussian noise and can be computationally intensive in high-dimensional settings.

**Detecting Causal Mechanism Shifts.** A broader line of research focuses on detecting the presence of a CMS without dissecting its cause. At the individual node level, this is treated as a two-sample testing problem for conditional distributions (Lee et al., 2024; Yan et al., 2022), addressed by various non-parametric tests and deep generative models (Kostic et al., 2024; Chang et al., 2024). However, many approaches, including recent conditional two-sample tests (Hu & Lei, 2024; Chen & Lei, 2025; Chatterjee et al., 2024), require per-node examination of the conditional distribution within the DAG, leading to high computational complexity.

Another relevant area is Intervention Target Estimation (ITE), which aims to efficiently identify which nodes in a causal graph have been affected by interventions. Early approaches in the linear and soft-intervention setting, such as CITE (Varici et al., 2021), exploit changes in the precision matrix to scalably recover targets without learning the full DAG. Follow-up work, such as PreDITEr (Varici et al., 2022), extends this framework to handle latent confounding, demonstrating that one can still characterize and estimate valid intervention targets even without full structural recovery. More recently, methods such as LIT (Yang et al., 2024) broaden the scope to heterogeneous and nonlinear environments, proposing a two-step recovery and matching procedure to pinpoint unknown targets. Deep learning-based methods, such as DeepITE (Tao et al., 2024), leverage graph autoencoders and auxiliary labels to jointly learn across multiple intervention instances, thereby improving scalability and generalization. Overall, while these methods have made significant progress in detecting intervention targets, they do not explicitly dissect the nature of these changes. In contrast, the central goal of our framework is to perform both detection and dissection simultaneously. Furthermore, our approach can analyze local mechanism shifts at specific values of parent variables, enabling a more focused and targeted investigation of where and how a shift has occurred.

## 2. Preliminaries

We are using the structural causal models (Pearl, 2009) as a framework for data generating mechanisms across environments: a base environment (1) and a new environment (2), where a causal mechanism shift has occurred. Given the data from both environments and the causal DAG from the base environment, our goal is to identify the nodes affected by the shift and classify their type as either a function shift or a noise shift.

**Definition 2.1** (SCM (Pearl, 2009)). A structural causal model (SCM) $\mathcal{M} = (f, P_\varepsilon)$ is composed of a collection of $J$ structural equations:

$$X_j := f_j(\text{PA}_j, \varepsilon_j), \quad j \in \{1, ..., J\}. \tag{1}$$

Here, $P_\varepsilon$ is a joint distribution over the noise variables, where $\varepsilon_i$ and $\varepsilon_j$ for $i, j \in \{1, ..., J\}$ are jointly independent. $\varepsilon_j$ is an exogenous variable and $\text{PA}_j$ denotes parent variables of node $X_j$, independent of $\varepsilon_j$. The structural equations are determined by a vector of functions $f = [f_1, ..., f_J]$. Since we assume the independence condition for noise variables, the set of observed variables is causally sufficient.

To leverage the framework of Causal Normalizing Flows (CNF) (Javaloy et al., 2023), we make the following assumptions.

**Assumption 2.2.** For each $j \in \{1, \ldots, J\}$, the function $f_j$ is a diffeomorphism; i.e., for any fixed realization of $\text{PA}_j = \text{pa}_j$, the function $f_j$ mapping from $\varepsilon_j$ to $X_j$ is bijective, and both the mapping and its inverse are differentiable.

**Definition 2.3** (Environment). Each environment $k \in \{1, 2\}$ is characterized by a data distribution $P_X^{(k)}$ generated by an SCM $\mathcal{M}^{(k)}$. We denote the random variable $X_j$ as observed in environment $k$ by $X_j^{(k)}$. Different environments $k \neq k'$ are distinguished by the fact that at least one causal mechanism in $\mathcal{M}^{(k)}$ differs from the corresponding one in $\mathcal{M}^{(k')}$.

Furthermore, for the application of our method, we introduce an assumption on the structure of the causal mechanisms:

**Assumption 2.4.** Each structural equation $f_j$ can be decomposed as:

$$f_j(\text{PA}_j, \varepsilon_j) = g_j(h_j(\text{PA}_j), \varepsilon_j), \tag{2}$$

where $h_j : \mathbb{R}^{d_j} \to \mathbb{R}^{k_j}$ is a function of the parents only, and $g_j : \mathbb{R}^{k_j} \times \mathbb{R} \to \mathbb{R}$ is a differentiable link function that combines their outputs. $g_j$ is a link function invariant across environments. Variations in $\text{PA}_j$ are absorbed solely through $h_j$, which means that changes in $\text{PA}_j$ are restricted to those that can be fully explained by modifications in $h_j$.

The SCM in each environment $\mathcal{M}^{(k)}$ is defined as follows:

$$X_j^{(k)} := f_j^{(k)}(\text{PA}_j^{(k)}, \varepsilon_j^{(k)}) = g_j(h_j^{(k)}(\text{PA}_j^{(k)}), \varepsilon_j^{(k)}). \tag{3}$$

**Definition 2.5** (Causal Mechanism Shift). (Chen et al., 2023; Mameche et al., 2024) A causal mechanism shift at $j$ refers to a situation where the conditional distributions of a variable $X_j$ given its parents differ between two environments:

$$P(X_j^{(1)}|\text{PA}_j^{(1)}) \neq P(X_j^{(2)}|\text{PA}_j^{(2)}). \tag{4}$$

For example, Figure 1 shows $X_2$ and $X_3$ experience causal mechanism shifts. Let the causal graph for environment 1 be $\mathcal{G}^{(1)}$. We assume that the causal structure in environment 2 results from interventions on $\mathcal{G}^{(1)}$. It implies that the parent set for each variable $X_j$ in environment 2 is a subset of the parent set for $X_j$ in environment 1.

**Assumption 2.6.** For a node $j$ that doesn't experience a shift, the conditional probabilities remain invariant.

$$P(X_j^{(1)}|\text{PA}_j^{(1)}) = P(X_j^{(2)}|\text{PA}_j^{(2)}).$$

**Definition 2.7** (Shifted Nodes). Indices $j$ for variables $X_j$ that experience a causal mechanism shift are referred to as shifted nodes, denoted by the set $S$:

$$S = \{1 \leq j \leq J : P(X_j^{(1)} \mid \text{PA}_j^{(1)}) \neq P(X_j^{(2)} \mid \text{PA}_j^{(2)})\}.$$

A shift across environments is characterized by $S \neq \varnothing$. The shift can be categorized into two types:

**Assumption 2.8** (Shift Types). Environment 2 differs from environment 1 in one of the following ways for a node $X_j$: a function shift ($h_j^{(1)} \neq h_j^{(2)}$) or a noise shift ($\varepsilon_j^{(1)} \overset{d}{\neq} \varepsilon_j^{(2)}$).

We use the notation $\overset{d}{=}$ and $\overset{d}{\neq}$ to denote equality and inequality in distribution, respectively. The function difference is defined as $\Delta h_j(\text{pa}) := h_j^{(2)}(\text{pa}) - h_j^{(1)}(\text{pa})$ for any given input value **pa**. A function shift covers cases where an incoming edge to $X_j^{(2)}$ is removed. This corresponds to the new function $h_j^{(2)}$ mapping the input from that parent to a constant value. Since a simultaneous shift of function and noise is often indistinguishable from a shift in the function alone, we analyze shifts as two separate cases for identifiability. Section 3.3 provides an example illustrating the non-identifiability and presents a strategy to make the shifts identifiable.

Function shifts and noise shifts are not mutually exclusive; i.e., some shifts may reside in the intersection between function and noise shifts. For example, consider the following observed shifts:

$$X_j^{(1)} = \text{PA}_j^{(1)}\varepsilon_j^{(1)}, \quad X_j^{(2)} = 2\text{PA}_j^{(2)}\varepsilon_j^{(2)}.$$

This shift can be viewed as a functional shift by transforming $\mathrm{PA}_j$ to $2\mathrm{PA}_j$; or a noise shift which transforms $\varepsilon_j$ to $2\varepsilon_j$. For such shifts that can be considered as function and noise shifts, throughout the paper, we will consider it as a noise-shift. Formally, let $\mathcal{T}_{g_j}$ be the set of functional variations $\delta_j : \mathbb{R}^{d_j} \to \mathbb{R}^{k_j}$ such that the composition with the link function can be factored into a noise transformation:

$$\mathcal{T}_{g_j} := \{\delta_j \mid \exists \psi : \mathbb{R} \to \mathbb{R} \text{ such that } \psi \text{ is invertible and}$$

$$\forall \mathbf{pa}, \ g_j(h_j^{(1)}(\mathbf{pa}) + \delta_j(\mathbf{pa}), \varepsilon_j^{(1)}) \stackrel{d}{=} g_j(h_j^{(1)}(\mathbf{pa}), \psi(\varepsilon_j^{(1)}))\}.$$

If a shift is categorized as a function shift ($h_j^{(2)} = h_j^{(1)} + \Delta h_j$), then the difference term satisfies $\Delta h_j \notin \mathcal{T}_{g_j}$. If $\Delta h_j \in \mathcal{T}_{g_j}$, we categorize it as a noise shift due to non-identifiability. As a caveat, we make sure that this shift is different from simultaneous shift such as

$$X_j^{(1)} = \mathrm{PA}_j^{(1)}\varepsilon_j^{(1)}, \quad X_j^{(2)} = \exp(\mathrm{PA}_j^{(2)})(2\varepsilon_j^{(1)}).$$

For any SCM that induces a DAG, we can express observed data by exogenous variables. We denote $X^{(k)} = [X_1^{(k)}, ..., X_J^{(k)}]$ and $\varepsilon^{(k)} = [\varepsilon_1^{(k)}, ..., \varepsilon_J^{(k)}]$. For environment 1, $X^{(1)}$ is expressed as follows:

$$\begin{aligned}
[X_1^{(1)}, ..., X_J^{(1)}] &= [f_1^{(1)}(\mathrm{PA}_1^{(1)}, \varepsilon_1^{(1)}), ..., f_J^{(1)}(\mathrm{PA}_J^{(1)}, \varepsilon_J^{(1)})] \\
&= [\tilde{f}_1^{(1)}(\varepsilon_{\mathrm{an}_1}^{(1)}), ..., \tilde{f}_J^{(1)}(\varepsilon_{\mathrm{an}_J}^{(1)})] \\
&= \tilde{f}^{(1)}([\varepsilon_1^{(1)}, ..., \varepsilon_J^{(1)}]) = \tilde{f}^{(1)}(\varepsilon^{(1)}).
\end{aligned}$$

Since all parent variables consist of noise variables, $\varepsilon_{\mathrm{an}_j}^{(1)}$ denotes the noise ancestors that determine $X_j^{(1)}$. Equivalently, by unrolling the recursive definitions (Javaloy et al., 2023) of each $f_j^{(1)}$, we obtain a non-recursive map $\tilde{f}^{(1)} : \mathbb{R}^J \to \mathbb{R}^J$ such that

$$X^{(1)} = \tilde{f}^{(1)}(\varepsilon^{(1)}). \tag{5}$$

**Definition 2.9** (Causal Normalizing Flow). (Javaloy et al., 2023) Causal Normalizing Flow (CNF) $T(X)$ is an autoregressive normalizing flow (ANF) (Kingma et al., 2016; Papamakarios et al., 2017) to learn an invertible transformation from i.i.d random variables $X = [X_1, ..., X_j]$ to a set of independent base-noise variables $N = [N_1, ..., N_J]$.

$$T(X) = T(\tilde{f}(\varepsilon)) = [N_1, ..., N_J]. \tag{6}$$

Base-noise variables are random variables that follow simple base distributions, such as the standard normal distribution. According to Theorem 1 in (Javaloy et al., 2023), for each $X_j \in X$, there exists an invertible function $q_j$ between true noise $\varepsilon_j$ from a structural causal model and base noise $N_j$ in a causal normalizing flow, where both generate the same observational data:

$$q_j(\varepsilon_j) = N_j, \quad \forall j \in \{1, ..., J\}. \tag{7}$$

In our framework, CNF is trained on the observed dataset $X^{(1)}$ from environment 1, denoted as $\widehat{T}^{(1)}$. We denote $\widehat{T}^{(1)}(X^{(1)})_j$ as the $j$-the element of $\widehat{T}^{(1)}(X^{(1)})$. The trained CNF yields the estimated noise $\widehat{N}^{(1)} = \widehat{T}^{(1)}(X^{(1)})$. $\widehat{T}^{(1)}$ is then applied to the data from environment 2; i.e., $\widehat{T}^{(1)}(X^{(2)})$.

## 3. Method

We assume that Assumptions 2.2-2.8 hold in this section. Our method leverages a single Causal Normalizing Flow (CNF) model, trained only on data from the base environment (environment 1), to perform a comprehensive analysis of causal mechanism shifts.

First, in Section 3.1, the trained CNF is used to estimate the conditional probability distributions for each node under both environments. By measuring the divergence between these distributions, we detect the set of shifted nodes, $S$. For the nodes in $S$, Section 3.2 details the dissection process. We input the data from environment 2 into the same CNF model to generate a residual noise, which captures the discrepancy between the original mechanism and the new data. A statistical test for independence between this residual noise and the parents in environment 2 then allows us to classify the cause: independence indicates a noise shift, while dependence implies a function shift. Finally, Section 3.3 addresses the more subtle challenge of non-identifiability between a function-only shift and a simultaneous function-and-noise shift, and we present a set of formal assumptions under which they become distinguishable.

While the existence of a true mapping $q : \varepsilon^{(1)} \to N^{(1)}$ is theoretically guaranteed by Theorem 1 in (Javaloy et al., 2023), in practice, it is estimated from finite samples. We denote the approximation of this mapping, implemented as a CNF $\widehat{T}^{(1)}$, by $\hat{q}$. We assume that $\hat{q}$ is invertible.

### 3.1. Detection of Shift

To detect a shift for a node $X_j$ between environments 1 and 2, we test the null hypothesis that their conditional probability distributions are identical:

$$H_{0,j} : \ P\left(X_j^{(1)} \mid \mathrm{PA}_j^{(1)}\right) = P\left(X_j^{(2)} \mid \mathrm{PA}_j^{(2)}\right)$$

To test this hypothesis, we leverage the model $\widehat{T}^{(1)}$. The core idea is to assess how well $\widehat{T}^{(1)}$ explains $X^{(2)}$ by simulating a counterfactual distribution on data from environment 1. This is based on the do-operator in CNF (Javaloy et al., 2023).

Let $P_{\mathrm{do}(\mathrm{PA}_j=c)}^{(1)}$ be the counterfactual distribution generated by applying the $do(\mathrm{PA}_j = c)$ to Environment 1 data using the model $\widehat{T}^{(1)}$. Let $\tilde{P}_{\mathrm{do}(\mathrm{PA}_j=c)}^{(2)}$ be the counterfactual

**FANS: Function And Noise Separation**

**INPUT**

**Dataset from Environment 1** $(X_1^{(1)}, X_2^{(1)}, X_3^{(1)})$    **New dataset from Environment 2** $(X_1^{(2)}, X_2^{(2)}, X_3^{(2)})$

**Causal DAG**    $X_1^{(1)} \longrightarrow X_2^{(1)} \longrightarrow X_3^{(1)}$

**ALGORITHM**

**(1) Train a Causal Normalizing Flow** $\widehat{T}^{(1)}(X^{(1)}) = N^{(1)}$

**(2) Calculate divergence between counterfactual distributions** $\mathcal{D}\left(P_{\text{do}(\text{PA}_j=c)}^{(1)}, \tilde{P}_{\text{do}(\text{PA}_j=c)}^{(2)}\right)$
The aggregated divergence exceeds a threshold: **Shifts Detected {2, 3}**

**(3) Perform independence tests** $H_0 : \widehat{T}^{(1)}(X^{(2)})_j \perp \text{PA}_j^{(2)}$    Reject the null hypothesis: **Function shift {2}**
Accept the null hypothesis: **Noise shift {3}**

*Figure 2.* Assuming the data follows the Structural Equation Model (SEM) described in Figure 1, the set of true shifted nodes is $\{2, 3\}$, where $X_2$ undergoes a function shift and $X_3$ undergoes a noise shift. This figure illustrates the required inputs and the algorithmic procedure of FANS to detect and dissect these shifts.

distribution generated by applying the same intervention procedure to Environment 2 data, also using the model $\widehat{T}^{(1)}$.

Let $c$ be a point in the support of the parent variables $\text{PA}_j^{(1)}$. The local divergence at point $c$ is defined as the divergence between the two local conditional distributions:

$$\mathcal{D}_{j,c} := \mathcal{D}\left(P_{\text{do}(\text{PA}_j=c)}^{(1)}, \tilde{P}_{\text{do}(\text{PA}_j=c)}^{(2)}\right) \tag{8}$$

where $\mathcal{D}$ is a divergence measure. $\mathcal{D}$ is any divergence satisfying $\mathcal{D}(P, Q) \geq 0$ and $\mathcal{D}(P, Q) = 0$ iff $P = Q$. Let the test statistic, $\bar{\mathcal{D}}_j$, be the empirical average of this local divergence over $n$ points $\{c_1, \dots, c_n\}$ sampled from the parent distribution $P(\text{PA}_j)$. As the number of samples $n$ approaches infinity, $\bar{\mathcal{D}}_j$ converges to the expected divergence:

$$\mathbb{E}_{c \sim P(\text{PA}_j^{(1)})}[\mathcal{D}_{j,c}]$$

**Theorem 3.1** (Shift Detection Equivalence). *A causal mechanism shift at node $X_j$ has occurred if and only if the expected divergence is strictly greater than zero.*

$$\mathbb{E}_{c \sim P(\text{PA}_j^{(1)})}[\mathcal{D}_{j,c}] > 0 \iff$$
$$P(X_j^{(1)}|\text{PA}_j^{(1)}) \neq P(X_j^{(2)}|\text{PA}_j^{(2)})$$

A node is identified as shifted if this aggregated value exceeds a predefined threshold $\tau$:

$$S = \{ j : \bar{\mathcal{D}}_j > \tau \}. \tag{9}$$

For the index $j \in S$, identified as shifted, we perform a dissection to determine whether the shift is primarily due to a function shift or a noise shift.

### 3.2. Dissection of Shift

To understand the type of shift, we start with a question: what noise would explain the new observation $X_j^{(2)}$ if we

apply the base model from environment 1? We refer to this hypothetical noise as the residual noise $\tilde{\varepsilon}_j^{(2)}$. It acts as a proxy that captures the discrepancy between the original mechanism and the new data.

$$X_j^{(2)} = g_j(h_j^{(1)}(\text{PA}_j^{(2)}), \tilde{\varepsilon}_j^{(2)}).$$

Due to our assumption that $f_j^{(1)}$ is invertible, the solution is uniquely determined by:

$$\tilde{\varepsilon}_j^{(2)} = [(\tilde{f}^{(1)})^{-1}(X^{(2)})]_j.$$

We infer noise from $X^{(2)}$ by $\widehat{T}^{(1)}$. The learned transformation $\widehat{T}^{(1)}$ maps the data to noise:

$$\widehat{T}^{(1)}(X^{(1)}) = \hat{q}(\varepsilon^{(1)}) = \hat{q}((\tilde{f}^{(1)})^{-1}(X^{(1)})).$$

By applying this transformation to the new data $X^{(2)}$, we obtain the transformed residual noise:

$$\widehat{T}^{(1)}(X^{(2)}) = \hat{q}\left((\tilde{f}^{(1)})^{-1}(X^{(2)})\right) = \hat{q}(\tilde{\varepsilon}^{(2)}). \tag{10}$$

**Theorem 3.2** (Independence Criterion). *For a shifted node $X_j$, the statistical independence of its residual noise $\tilde{\varepsilon}_j^{(2)}$ and its parents $\text{PA}_j^{(2)}$ indicates the type of shift:*

$$\tilde{\varepsilon}_j^{(2)} \not\perp \text{PA}_j^{(2)} \iff \textit{the shift is a function shift.}$$
$$\tilde{\varepsilon}_j^{(2)} \perp \text{PA}_j^{(2)} \iff \textit{the shift is a noise shift.}$$

A fundamental property of statistical independence is that it is preserved under transformations like $\hat{q}$. The conclusion of Theorem 3.2 holds for both the original and transformed noise.

$$\tilde{\varepsilon}_j^{(2)} \perp \text{PA}_j^{(2)} \iff \hat{q}_j(\tilde{\varepsilon}_j^{(2)}) \perp \text{PA}_j^{(2)}.$$

**Algorithm 1** FANS: Function And Noise Separation for Dissecting Causal Mechanism Shifts

1: **Input:** $X^{(1)}, X^{(2)}$, and $\mathcal{G}^{(1)}$.
2: **Output:** $\widehat{S}$, and for each $j \in \widehat{S}$, the identified shift type (Function, Noise).
3: **Phase 1: Learning the Base Causal Mechanism**
4: Train CNF $\widehat{T}^{(1)}$ on $X^{(1)}$.
5: **Phase 2: Shift Detection**
6: Initialize $\widehat{S} \leftarrow \emptyset$.
7: **for** each node index $j$ **do**
8:    Evaluate the divergence $\mathcal{D}_{j,c}$      (Eq. 8)
9:    Aggregate over $c$ to compute $\bar{\mathcal{D}}_j$
10:   **if** $\bar{\mathcal{D}}_j > \tau$ **then**
11:      add $j$ to the set of shifted nodes: $\widehat{S} \leftarrow \widehat{S} \cup \{j\}$.
12:   **end if**
13: **end for**
14: **Phase 3: Shift Dissection**
15: **for** each index for shifted nodes $j \in \widehat{S}$ **do**
16:   $\hat{q}_j(\tilde{\varepsilon}_j^{(2)}) \leftarrow \widehat{T}^{(1)}(X^{(2)})_j$     (Eq. 10)
17:   Perform a statistical independence test for:

$$H_{0,j}^{\text{indep}} : \hat{q}_j(\tilde{\varepsilon}_j^{(2)}) \perp \text{PA}_j^{(2)}$$

18:   **if** $H_{0,j}^{\text{indep}}$ is rejected at level $\alpha$ **then**
19:      Label the shift in $j$ as a **Function Shift**.
20:   **else**
21:      Label the shift in $j$ as a **Noise Shift**.
22:   **end if**
23: **end for**
24: **return** $\widehat{S}$ and the labels for each node in $\widehat{S}$.

This allows us to dissect the shift type by analyzing the independence properties of the transformed noise obtained from the CNF.

To test $H_{0,j}^{\text{indep}}$, we use distance correlation as the test statistic to quantify the degree of dependence. We calculate the distance correlation between $\hat{q}_j(\tilde{\varepsilon}_j^{(2)})$ and $\text{PA}_j^{(2)}$, and reject the test hypothesis if this distance correlation exceeds a certain threshold. Permutation tests based on distance covariances are also applicable.

### 3.3. Non-Identifiability between Function-Only and Simultaneous Shifts

A key challenge arises when a shift involves the function, as a function-only shift can be observationally indistinguishable from a simultaneous shift of both function and noise. The following example illustrates the non-identifiability.

We consider a structural equation $X_j = g_j(h_j(\text{PA}_j), \varepsilon_j)$ with an additive link function $g_j(h_j(\text{PA}_j), \varepsilon) = a_1 + a_2 \cdot \varepsilon$, where $h_j(\text{PA}_j) = [a_1, a_2]^\top$. Let $\sigma(\cdot)$ denote the sigmoid

function to ensure strict invertibility ($a_2 > 0$). Our base environment is as follows:

$$X_j^{(1)} = \text{PA}_j^{(1)} + \sigma(\text{PA}_j^{(1)}) \cdot \varepsilon_j^{(1)}, \ \varepsilon_j^{(1)} \sim U(0, 1).$$

Here, $U(\cdot, \cdot)$ denotes a uniform distribution. Data comes from two environments:

(Case 1) $h_j^{(2)}(\text{PA}_j) = [\sigma(\text{PA}_j^{(2)}), \ \sigma(\text{PA}_j^{(2)})]^\top$

$$X_j^{(2)} = \sigma(\text{PA}_j^{(2)}) + \sigma(\text{PA}_j^{(2)})\varepsilon_j^{(2)}, \ \varepsilon_j^{(2)} \sim U(0, 1)$$

(Case 2) $h_j^{(2)}(\text{PA}_j) = [1.5\sigma(\text{PA}_j^{(2)}), \ \sigma(\text{PA}_j^{(2)})]^\top$

$$X_j^{(2)} = 1.5\sigma(\text{PA}_j^{(2)}) + \sigma(\text{PA}_j^{(2)})\varepsilon_j^{(2)}, \ \varepsilon_j^{(2)} \sim U(-\frac{1}{2}, \frac{1}{2})$$

Both cases yield the same conditional distribution $X_j^{(2)}|\text{PA}_j^{(2)} \sim U(\sigma(\text{PA}_j^{(2)}), 2\sigma(\text{PA}_j^{(2)}))$. Since the scale term is strictly positive, the invertibility assumption (Assumption 2.2) holds in both scenarios. However, the data alone cannot distinguish whether the shift originated solely from the function parameters (Case 1) or from a combination of function scaling and noise reduction (Case 2). We find that if Assumption 3.3 is satisfied, function shifts and function-noise shifts become identifiable.

**Assumption 3.3** (Generalized Affine Interaction)**.** Let $g_j^*(h_j^{(1)}, N_j) := g_j(h_j^{(1)}, q_j^{-1}(N_j)) = g_j(h_j^{(1)}, \varepsilon_j)$. We assume there exist real-valued functions $\beta_{j,0}(h)$ and $\beta_{j,1}(h)$ such that the following relationship holds for the function shift:

$$g_j^*(h_j, N_j) = \Psi_j(\beta_{j,1}(h_j) \cdot N_j + \beta_{j,0}(h_j)).$$

$\Psi_j : \mathbb{R} \mapsto \mathbb{R}$ is an invertible and differentiable scalar function that wraps the affine combination. This assumption posits that the function $g_j^*$ exhibits a specific structure. Under the assumptions, we can isolate the effect of the function shift $\Delta h$ in $q_j(\tilde{\varepsilon}_j^{(2)})$. After standardizing the noise by removing conditional moments, the resulting distribution is shown to be independent of the parents $\text{PA}_j^{(2)}$. This leads to the following theorem.

**Theorem 3.4** (Shift Identifiability)**.** *Let $\tilde{q}_j(\tilde{\varepsilon}_j^{(2)})$ be the standardized inferred noise:*

$$\tilde{q}_j(\tilde{\varepsilon}_j^{(2)}) = \frac{q_j(\tilde{\varepsilon}_j^{(2)}) - \mathbb{E}[q_j(\tilde{\varepsilon}_j^{(2)})|\text{PA}_j^{(2)}]}{\sqrt{\text{Var}(q_j(\tilde{\varepsilon}_j^{(2)})|\text{PA}_j^{(2)})}}.$$

*Under Assumption 3.3 and assuming the CNF base noise follows a standard normal distribution, if the causal mechanism shift is function-only, then the squared standardized noise follows a Chi-squared distribution with one degree of freedom.*

$$(\tilde{q}_j(\tilde{\varepsilon}_j^{(2)}))^2 \sim \chi^2(1).$$

The converse holds when a function-noise shift causes $q_j(\varepsilon_j^{(2)})$ to deviate from a normal distribution. The full derivation is provided in the appendix, and Algorithm 1 presents the pseudo-code of FANS.

### 3.4. Finite-Sample Consistency of the Dissection Test

In finite-sample settings, the estimated CNF map $\hat{q}$ introduces approximation errors, resulting in a non-zero baseline dependence. To address a non-zero baseline bias, our empirical statistic $\Delta\hat{\mathcal{R}}$ computes the absolute difference in distance correlations between the two environments. We show that, under standard moment conditions, the empirical dissection statistic is consistent: function and noise shifts are correctly classified with probability tending to one as the sample size grows.

Let $Z^{(k)} := \mathrm{PA}_j^{(k)}$ denote the parents in environment $k$, and let $\mathcal{R}(\cdot,\cdot)$ denote the distance correlation (Székely et al., 2007). Define

$$\Delta\mathcal{R}^* := |\mathcal{R}(N^{(2)}, Z^{(2)}) - \mathcal{R}(N^{(1)}, Z^{(1)})|, \qquad (11)$$

$$\Delta\hat{\mathcal{R}} := |\mathcal{R}(\hat{N}^{(2)}, Z^{(2)}) - \mathcal{R}(\hat{N}^{(1)}, Z^{(1)})|. \qquad (12)$$

By Theorem 3.2, $\Delta\mathcal{R}^* = 0$ under a pure noise shift and $\Delta\mathcal{R}^* > 0$ under a function shift.

**Assumption 3.5** ($L^2$ Estimation Consistency). *For each $j$ and $k \in \{1, 2\}$, $\mathbb{E}[(\hat{N}_j^{(k)} - N_j^{(k)})^2] \le \delta_n^2$, with $\delta_n \to 0$ as $n \to \infty$.*

**Assumption 3.6** (Finite Second Moments). *$\mathbb{E}[(N_j^{(k)})^2] < \infty$ and $\mathbb{E}[\|Z^{(k)}\|^2] < \infty$.*

**Proposition 3.7** (Consistency of the Dissection Statistic). *Under Assumptions 3.5 and 3.6,*

$$\Delta\hat{\mathcal{R}} \xrightarrow{p} \Delta\mathcal{R}^* \quad as \; n \to \infty. \qquad (13)$$

*Proof sketch.* Cauchy–Schwarz bounds the distance covariance error as $|\mathcal{V}^2(\hat{N}, Z) - \mathcal{V}^2(N, Z)| \le 8C_Z\delta_n$, where $C_Z := \sqrt{\mathbb{E}[\|Z - Z'\|^2]}$. The continuous mapping theorem, applied to the distance correlation functional, then yields convergence in probability of $\Delta\hat{\mathcal{R}}$ to $\Delta\mathcal{R}^*$. Full details are provided in Appendix B.5.

**Corollary 3.8** (Consistent Classification). *Fix any threshold $\tau \in (0, \Delta\mathcal{R}^*)$ when $\Delta\mathcal{R}^* > 0$. Under Assumptions 3.5 and 3.6:*

- *If the shift is a pure noise shift ($\Delta\mathcal{R}^* = 0$), then $\Pr(\Delta\hat{\mathcal{R}} > \tau) \to 0$ as $n \to \infty$.*

- *If the shift is a function shift ($\Delta\mathcal{R}^* > 0$), then $\Pr(\Delta\hat{\mathcal{R}} > \tau) \to 1$ as $n \to \infty$.*

Proposition 3.7 complements the population-level identifiability of Theorem 3.2 with a finite-sample guarantee: as the

CNF training set grows, the probability of misclassifying a noise shift as a function shift, or vice versa, vanishes. This provides theoretical grounding for the empirical robustness observed across sample sizes and model capacities in our experiments.

## 4. Experiments

### 4.1. Synthetic data

We evaluate the model's ability to detect causal mechanism shifts in Table 1. Baseline methods for comparison include iSCAN (Chen et al., 2023), LinearCCP (Huang et al., 2024), LCIT (Duong & Nguyen, 2022), PreDITEr (Varici et al., 2022) and SplitKCI (Pogodin et al., 2024). The data generation process follows (Chen et al., 2023). Initially, 30 random DAGs were generated for each of two graph types: Erdős-Rényi (ER) and Scale-Free (SF). To be specific, the average number of edges is equal to $4J$. We also randomly selected $0.2J$ shifted nodes. The number of nodes $J$ within each DAG was set to 10, 20, 30, 40, and 50.

We generated 50,000 data points for each case. Some baseline methods were evaluated on a 1,000 sample subset due to computational constraints. For the detection task, iSCAN and SplitKCI were limited by memory complexity. LCIT was limited by time complexity. For the dissection task, GPR was restricted by time complexity. iSCAN was also limited to 1,000 samples due to excessive Type I error. The subsequent residual analysis step of FANS was also performed on this 1,000 sample subset. A Causal Mechanism Shift (CMS) was introduced between environments 1 and 2. The SCM for environment 1 is defined as:

$$X_j^{(1)} := \sum_k \sin([\mathrm{PA}_j^{(1)}]_k^2) + \sigma\Big(\sum_k ([\mathrm{PA}_j^{(1)}]_k^2)\Big) \cdot \varepsilon_j^{(1)}.$$

Here, $\sigma(\cdot)$ denotes a sigmoid function and $[\mathrm{PA}_j^{(1)}]_k$ denotes $k$-th element of the parents $\mathrm{PA}_j^{(1)}$ with $\varepsilon_j \sim \mathcal{N}(0, 1)$. A function shift can manifest in two distinct ways. The first is a change from a $\sin(\cdot)$ function to a $\cos(2(\cdot)^2 - 3(\cdot))$ function. The second is a shift where the set of causal parents is altered by the deletion of a parent. Separately, a noise shift occurs when the distribution of the exogenous noise term is modified. This can take the form of a variance change, such as $\varepsilon_j \sim \mathcal{N}(0, 2)$, or a change in the distribution's family to $\varepsilon_j \sim \mathrm{Laplace}(0, 1)$.

Each node $j$ in the graph can only exhibit one of three cases: no shift, a function shift, or a noise shift. Table 1 evaluates the model's ability to identify which nodes have undergone any type of shift. Table 1 indicates that the FANS framework is effective for the task of shifted node detection, outperforming established baseline methods in the tested configurations. Its robust performance, particularly on SF graphs, underscores its potential utility in analyzing CMS

Table 1. F1-scores for **Shifted Nodes Detection**, detailed by graph type and node count. Results are shown across varying node counts (J) for Erdős-Rényi (ER) and Scale-Free (SF) graphs. Higher F1-scores (F1 ↑) indicate better performance. Our proposed method consistently outperforms all baseline methods across all experimental settings. Numbers in parentheses indicate standard deviations.

| Method | ER Graph - Nodes (J) | | | | | SF Graph - Nodes (J) | | | | |
|---|---|---|---|---|---|---|---|---|---|---|
| | 10 | 20 | 30 | 40 | 50 | 10 | 20 | 30 | 40 | 50 |
| iSCAN | 0.371 (0.327) | 0.484 (0.245) | 0.552 (0.149) | 0.527 (0.077) | 0.553 (0.117) | 0.321 (0.330) | 0.546 (0.233) | 0.357 (0.197) | 0.433 (0.170) | 0.478 (0.157) |
| SplitKCI | 0.428 (0.248) | 0.406 (0.173) | 0.431 (0.141) | 0.402 (0.118) | 0.417 (0.100) | 0.295 (0.227) | 0.301 (0.108) | 0.293 (0.097) | 0.320 (0.086) | 0.301 (0.067) |
| LCIT | 0.468 (0.199) | 0.468 (0.149) | 0.500 (0.085) | 0.537 (0.063) | 0.519 (0.069) | 0.353 (0.039) | 0.338 (0.026) | 0.342 (0.015) | 0.328 (0.020) | 0.328 (0.023) |
| LinearCCP | 0.731 (0.153) | 0.784 (0.117) | 0.753 (0.102) | 0.765 (0.089) | 0.772 (0.077) | 0.719 (0.172) | 0.714 (0.149) | 0.661 (0.134) | 0.623 (0.106) | 0.609 (0.121) |
| PreDITEr | 0.940 (0.115) | 0.978 (0.050) | 0.954 (0.077) | 0.965 (0.057) | 0.963 (0.040) | 0.933 (0.136) | 0.976 (0.054) | 0.959 (0.073) | 0.947 (0.067) | 0.972 (0.049) |
| **FANS (Ours)** | **0.989** (0.061) | **1.000** (0.000) | **0.980** (0.043) | **0.982** (0.030) | **0.975** (0.037) | **1.000** (0.000) | **1.000** (0.000) | **1.000** (0.00) | **1.000** (0.000) | **1.000** (0.000) |

Table 2. F1-scores for **Shift Type Dissection** across ER/SF graphs and node counts (J).

| Method | ER Graph - Nodes (J) | | | | | SF Graph - Nodes (J) | | | | |
|---|---|---|---|---|---|---|---|---|---|---|
| | 10 | 20 | 30 | 40 | 50 | 10 | 20 | 30 | 40 | 50 |
| iSCAN | 0.611 (0.351) | 0.687 (0.250) | 0.678 (0.224) | 0.645 (0.236) | 0.683 (0.198) | 0.722 (0.304) | 0.620 (0.297) | 0.664 (0.243) | 0.664 (0.231) | 0.580 (0.247) |
| LinearCCP | 0.700 (0.365) | 0.733 (0.195) | 0.749 (0.187) | 0.762 (0.174) | 0.751 (0.163) | 0.822 (0.300) | 0.823 (0.198) | 0.830 (0.211) | 0.799 (0.160) | 0.815 (0.161) |
| GPR | 0.911 (0.213) | 0.845 (0.230) | 0.888 (0.195) | 0.922 (0.126) | 0.906 (0.115) | 0.722 (0.304) | 0.722 (0.304) | 0.722 (0.304) | 0.726 (0.304) | 0.722 (0.304) |
| **FANS (Ours)** | **1.000** (0.000) | **0.986** (0.054) | **0.962** (0.088) | **0.968** (0.064) | **0.947** (0.068) | **1.000** (0.000) | **1.000** (0.000) | **1.000** (0.000) | **1.000** (0.000) | **1.000** (0.000) |

in diverse network environments. Table 2 is the result from the task of dissecting the function and noise. Baselines include iSCAN, LinearCCP, and GPR (Li et al., 2023). The results from Table 2 underscore the strength of the proposed FANS method, particularly in the combined detection and dissection task on smaller graphs and in pure shift type separation on ER graphs.

We further analyze the robustness of FANS through a series of additional experiments. FANS is robust to training hyperparameters: F1 scores remain stable across batch sizes and learning rates (Appendix D.1.1), and performance degrades gracefully under reduced sample sizes (down to $N = 5,000$), shorter training (down to 100 epochs), and detection/dissection threshold in Appendix D.1.2–D.2.2. Model capacity, in contrast, has a more pronounced effect: reducing the hidden dimensions from $[32, 32, 32]$ to $[8, 8]$ causes a substantial drop in both detection and dissection, underscoring the need for sufficient CNF expressiveness to capture non-linear, non-additive mechanisms (Appendix D.2.3). We further analyze FANS under assumption violations—including quadratic-noise violations of Assumption 3.3, the ANM special case, environment swapping, and unobserved confounding in Appendix D.3.

## 4.2. Simultaneous shift

In Section 3.3, we propose a method to dissect function-only and simultaneous shifts. Our proposed synthetic setting already satisfies Assumption 3.3. We generate two distinct sets of shifts: (1) function-only shifts and (2) simultaneous function and noise shifts. Function-only shifts altered the causal function from $\sin(\cdot^2)$ to $\cos(2(\cdot)^2 - 3(\cdot))$. Simultaneous shifts included the same function alteration but also changed the noise distribution to $\text{Laplace}(0, 1)$. To classify a given shift, we followed the procedure outlined by Theorem 3.4.

We used a Generalized Additive Model (GAM) to estimate the conditional mean and conditional variance. Using these estimates, we standardized the noise for each data point. The primary challenge was accurately estimating the conditional mean and variance using GAMs, resulting in moderate performance. Table 3 shows that our proposed FANS achieved a higher F1 score, indicating a superior ability

Table 3. F1-scores for distinguishing between Function-Only and Simultaneous shifts. Our method leverages Theorem 3.4.

| Method | F1-score (Macro) |
|---|---|
| GPR | 0.378 |
| **FANS (Ours)** | **0.622** |

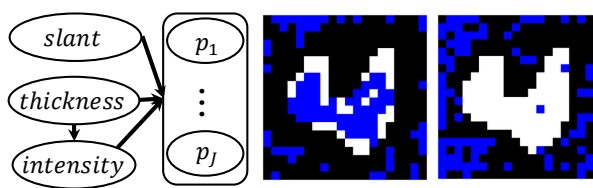

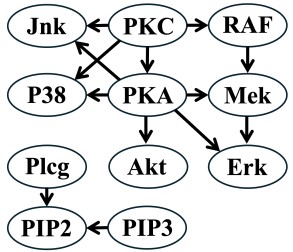

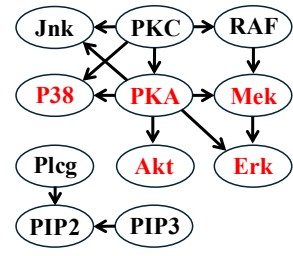

*Figure 3.* (Left) DAG of Morpho-MNIST, where slant, thickness, and intensity causally influence all pixel values, assuming conditional independence (Taylor-Melanson et al., 2024). (Middle) Top 100 shifted pixels (highlighted in blue) detected by FANS under the thickness function change. (Right) iSCAN results show weaker and less coherent sensitivity, confirming FANS's superior localization of function shifts. The white pixels represent the top-100 pixels with the largest pixel-wise differences between the mean images of each environment.

*(a)* The established causal graph of the protein signaling pathway (Kleinegesse et al., 2022)

*(b)* Nodes identified by FANS as having undergone a function shift are highlighted in red.

*Figure 4.* Application of FANS to the Sachs protein signaling dataset. The analysis identifies nodes affected by an intervention inhibiting the Mek protein (environment 2) compared to environment 1.

to distinguish between function-only and simultaneous shifts. The moderate F1 score reflects the inherent difficulty of this task. In Appendix D.3.1, we further examine FANS's behavior when Assumption 3.3 is violated by quadratic noise terms, reflecting the theoretical reliance of Theorem 3.4 on the generalized affine structure.

### 4.3. Morpho-MNIST

We evaluated our method on the Morpho-MNIST dataset to investigate the influence of stroke thickness on pixel values, using only digit 4. We created two distinct environments by first normalizing thickness to $[-1, 1]$ and then assigning thin digits (thickness $\leq -0.5$) to environment 1 and thicker digits (thickness $\geq 0$) to environment 2. To simulate a function shift, we then standardize the thickness range within each environment, transforming the underlying function as $f(x) \to f((x - a)/b)$. Our method can quantify the magnitude of a detected shift. This allows us to rank all pixels based on their degree of change in Figure 3. To quantify the localization accuracy, we computed the pixel-wise difference between the empirical averages of each environment and identified the top 100 pixels with the largest ground-truth shifts. Among these, FANS correctly detected 57 pixels, whereas iSCAN only detected 3 pixels.

### 4.4. Real-world protein signal data

The Sachs dataset comprises flow cytometry measurements of 11 proteins and phospholipids in the human T-cell signaling pathway under various experimental perturbations. Environment 1 served as the baseline using data from general T-cell activation, while Environment 2 contained intervention data where the Mek protein was inhibited with U0126. Although a definitive ground-truth DAG for this pathway remains debated, we adopted the established network from Sachs as the underlying causal structure. Our goal was to

demonstrate which nodes experience a shift in their generative mechanisms when U0126 is introduced. Our results are consistent with the intervention targets identified by other established methods. Figure 4 visualizes these results, showing that the consensus network (Mooij et al., 2020) points to Erk as an intervention node, FCI-JCI123 identifies Mek, Erk, and PKC, while another method (Eaton & Murphy, 2007) detects RAF, Mek, Erk, Akt, PKC, and JnK. Since an intervention on a node constitutes a function shift, our findings align with those from the Eaton and Murphy method. Notably, our method jointly identifies Mek, Erk, and Akt which is consistent with the findings of Eaton and Murphy.

## 5. Conclusion

We introduced FANS, a unified framework for detecting and dissecting causal mechanism shifts into function and noise components. By leveraging CNF, FANS extends dissection beyond additive noise models to general non-linear, non-additive settings, grounded in a theoretical independence criterion and a formal assumption that restores identifiability under simultaneous shifts. Empirical results on synthetic and real-world datasets corroborate our contributions.

**Limitations and future work.** FANS assumes a known causal DAG in the base environment and invertible structural equations, excluding discrete mechanisms and settings with unobserved confounding. The identifiability guarantee for simultaneous shifts further relies on the generalized affine interaction structure, which holds for common location-scale models but whose empirical scope merits further study. Promising directions include extending FANS to discrete variables via dequantization, integrating it with causal discovery to relax the known-DAG assumption, and leveraging its residual patterns as a diagnostic tool for hidden confounders.

## Impact Statement

This paper presents work whose goal is to advance the field of machine learning. There are many potential societal consequences of our work, none of which we feel must be specifically highlighted here.

## Acknowledgements

This work was supported by the National Research Foundation of Korea (NRF) grant funded by the Korea government (MSIT) (RS-2024-00457216), and the Institute of Information & communications Technology Planning & Evaluation (IITP) under the Leading Generative AI Human Resources Development (IITP-2026-RS-2026-25544647) grant funded by the Korea government (MSIT).

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

# A. Notation and Preliminaries

## A.1. Notation

| Variables and Metrics | |
| --- | --- |
| $X$ | Random vector $X = [X_1, \ldots, X_J]$, where $X_j$ denotes the $j$-th node |
| $\mathrm{PA}_j$ | Set of parent variables of node $X_j$ in the DAG |
| $\varepsilon_j$ | Exogenous noise variable corresponding to node $X_j$ |
| $k$ | Environment index, $k \in \{1, 2\}$ |
| $d_j$ | Dimension of the parent set $\mathrm{PA}_j$ (number of parent variables of $X_j$) |
| $k_j$ | Output dimension of the mapping $h_j$ |
| $\Delta h_j$ | Functional shift defined as $h_j^{(2)} - h_j^{(1)}$ |
| $N_j$ | Base noise variable in CNF (e.g., standard normal distribution) |
| $\varepsilon^{(k)}$ | Exogenous noise vector in environment $k$ |
| $\tilde{\varepsilon}^{(2)}$ | Inferred noise in environment 2 (shifted version of $\varepsilon^{(1)}$) |
| $x$ | Observed data point in $\mathbb{R}^J$ |
| $P_{\mathrm{do(PA}_j=c)}^{(1)}, \tilde{P}_{\mathrm{do(PA}_j=c)}^{(2)}$ | Counterfactual distributions in environments 1 and 2 |
| $\mathcal{D}_{j,c}$ | Divergence metric $\mathcal{D}\big(P_{\mathrm{do(PA}_j=c)}^{(1)} \,\|\, \widetilde{P}_{\mathrm{do(PA}_j=c)}^{(2)}\big)$ |
| $\bar{\mathcal{D}}_j^{\mathrm{FANS}}$ | Mean JSD over quantile conditioning points |
| $n, J$ | Sample size and number of nodes, respectively |
| $X_{\mathrm{do(PA}_j=c)}^{(1)}$ | Counterfactual variable in environment 1 |
| $\tilde{X}_{\mathrm{do(PA}_j=c)}^{(2)}$ | Counterfactual variable in environment 2 (mapped via env. 1 CNF) |
| **Functions** | |
| $f_j$ | A structural equation for node $j$ which satisfies diffeomorphism |
| $f_j^{(k)}$ | A structural equation for node $j$ in environment $k$ |
| $f^{(k)}$ | Collection of structural equations in environment $k$ |
| $\tilde{f}^{(k)}$ | Non-recursive mapping obtained by unrolling the SCM in environment $k$ such that $X^{(k)} = \tilde{f}^{(k)}(\varepsilon^{(k)})$ |
| $h_j$ | Mapping of parent variables $\mathbb{R}^{d_j} \to \mathbb{R}^{k_j}$ |
| $h_j^{(k)}$ | Parent feature mappings in environments 1 and 2 |
| $g_j$ | Environment-invariant differentiable link function $\mathbb{R}^{k_j} \times \mathbb{R} \to \mathbb{R}$ |
| $q_j$ | Invertible mapping between true noise $\varepsilon_j$ and base noise $N_j$ |
| $q(\cdot)$ | True invertible function mapping SCM noises to CNF base noises |
| $\hat{q}$ | Estimated invertible transformation of $q(\cdot)$ |
| $\widehat{T}^{(1)}, (\widehat{T}^{(1)})^{-1}$ | Learned CNF and inverse CNF for environment 1 |
| $\mathcal{I}(x, j, c)$ | Function computing the modified noise vector under $do(\mathrm{PA}_j = c)$ |

## A.2. Preliminaries

**Definition A.1** (Causal Normalizing Flow). (Javaloy et al., 2023) Causal Normalizing Flow (CNF) $T(X)$ is an autoregressive normalizing flow (ANF) (Kingma et al., 2016; Papamakarios et al., 2017) to learn an invertible transformation from i.i.d observed data $X$ to a set of independent base-noise variables $N = [N_1, ..., N_J]$.

$$T(X) = T(\tilde{f}(\varepsilon)) = [N_1, ..., N_J]. \tag{14}$$

Base noise variables $N$ are random variables that follow simple base distributions, such as the standard normal distribution. A CNF learns a mapping from observed data $X$ to these base noise variables $N$. Once the CNF is trained, we can map $X$ to $N$ or vice versa due to its invertibility. Theorem 1 in (Javaloy et al., 2023) states that if a true SCM and a CNF match, meaning both data generating processes yield the same distribution, the difference between the two mechanisms lies in an invertible function $q_j$. Specifically, $q_j$ takes the true exogenous noise of the SCM as input and outputs the corresponding CNF base noise variable $N_j$.

$$q_j(\varepsilon_j) = N_j, \quad \forall j \in \{1, ..., J\} \tag{15}$$

## B. PROOFS

### B.1. Counterfactual distribution in CNF

Let $\varepsilon^{(1)} \sim P_{\varepsilon^{(1)}}$, $\tilde{\varepsilon}^{(2)} \sim P_{\tilde{\varepsilon}^{(2)}}$. Let $\mathcal{I}(x, j, c)$ be the function that computes the modified noise vector required to perform $do(\text{PA}_j = c)$ on an initial data point $x$. We denote the parent indices as $j^*$. The procedure is defined as:

1. Infer initial noise: $\hat{u} = \widehat{T}^{(1)}(x)$.

2. Create an intervened data vector by substitution: $\text{pa}_j \leftarrow c = (x_1, \ldots, c, \ldots, x_J)$, where the parent of $j$th node is replaced by $c$.

3. Recalculate the noise component: $u_{\text{PA}_j \leftarrow c} = \left( \widehat{T}^{(1)}(\text{pa}_j \leftarrow c) \right)_{j^*}$.

4. Assemble the final modified noise vector: $\hat{u}_{do(\text{PA}_j=c)} = (\hat{u}_1, \ldots, u_{\text{PA}_j \leftarrow c}, \ldots, \hat{u}_J)$.

Thus, $\mathcal{I}(x, j, c) = \hat{u}_{do(\text{PA}_j=c)}$. The counterfactual random variable is defined as

$$X^{(1)}_{do(\text{PA}_j=c)} = (\widehat{T}^{(1)})^{-1}(\mathcal{I}(X^{(1)}, j, c)), \quad \text{where } X^{(1)}_{do(\text{PA}_j=c)} \sim P^{(1)}_{do(\text{PA}_j=c)}.$$

Similarly,

$$\tilde{X}^{(2)}_{do(\text{PA}_j=c)} = (\widehat{T}^{(1)})^{-1}(\mathcal{I}(X^{(2)}, j, c)), \quad \text{where } X^{(2)}_{do(\text{PA}_j=c)} \sim \tilde{P}^{(2)}_{do(\text{PA}_j=c)}.$$

The distribution of $\tilde{X}^{(2)}_{do(\text{PA}_j=c)}$ is denoted as $\tilde{P}^{(2)}_{do(\text{PA}_j=c)}$. The test statistic for detecting a causal mechanism shift is defined as

$$\mathcal{D}_{j,c} := \mathcal{D}\big(P^{(1)}_{do(\text{PA}_j=c)} \,\|\, \tilde{P}^{(2)}_{do(\text{PA}_j=c)}\big).$$

### B.2. Proof of Theorem 3.1

**Theorem 3.1** A causal mechanism shift at node $X_j$ has occurred if and only if the expected divergence is strictly greater than zero.

$$P\big(X^{(1)}_j \mid \text{PA}^{(1)}_j\big) \neq P\big(X^{(2)}_j \mid \text{PA}^{(2)}_j\big) \quad \Longleftrightarrow \quad \mathbb{E}_{c \sim P(\text{PA}^{(1)}_j)}[\mathcal{D}_{j,c}] > 0,$$

where $\mathcal{D}_{j,c} := \mathcal{D}\big(P^{(1)}_{do(\text{PA}_j=c)} \,\|\, \widetilde{P}^{(2)}_{do(\text{PA}_j=c)}\big)$ and the counterfactual laws are defined via the CNF map $\mathcal{I}$ as below.

**Interventional pushforward defined via CNF.** Let $\widehat{T}^{(1)} : \mathbb{R}^J \to \mathbb{R}^J$ be the learned invertible CNF for Environment 1 with inverse $(\widehat{T}^{(1)})^{-1}$, and write

$$U^{(1)} = \widehat{T}^{(1)}(X^{(1)}) \sim P_{U^{(1)}}, \qquad \widehat{U}^{(2)} = \widehat{T}^{(1)}(X^{(2)}) \sim P_{\widehat{U}^{(2)}}.$$

Given $x$, target $j$, and value $c$, define the modified-noise map

$$\mathcal{I}(x, j, c) = \big(\hat{u}_1, \ldots, \hat{u}_{j-1}, (\widehat{T}^{(1)}(\text{pa}_j \leftarrow c))_{j^*}, \hat{u}_{j+1}, \ldots, \hat{u}_J\big), \qquad \hat{u} = \widehat{T}^{(1)}(x).$$

Set $G_{j,c} := (\widehat{T}^{(1)})^{-1} \circ \mathcal{I}(\cdot, j, c)$. Then

$$X^{(1)}_{do(\text{PA}_j=c)} := G_{j,c}(X^{(1)}) \sim P^{(1)}_{do(\text{PA}_j=c)}, \qquad \tilde{X}^{(2)}_{do(\text{PA}_j=c)} := G_{j,c}(X^{(2)}) \sim \widetilde{P}^{(2)}_{do(\text{PA}_j=c)}.$$

Assume absolute continuity of all laws and that $\mathcal{D}$ is nonnegative and equals 0 iff its arguments coincide.

**Lemma (Jacobian cancels)**  Let $x_{inv} = G_{j,c}^{-1}(x)$ and $u_{inv} = \widehat{T}^{(1)}(x_{inv})$. Then,

$$\tilde{p}_{\text{do}(\text{PA}_j=c)}^{(2)}(x) = p_{\text{do}(\text{PA}_j=c)}^{(1)}(x) \cdot \mathcal{C}(x), \qquad \mathcal{C}(x) := \frac{p_{\widehat{U}^{(2)}}(u_{inv})}{p_{U^{(1)}}(u_{inv})}.$$

*Proof of lemma.* By change of variables under the common diffeomorphism $G_{j,c}$, and setting $x_{inv} = G_{j,c}^{-1}(x)$:

$$p_{\text{do}(\text{PA}_j=c)}^{(1)}(x) = p_{X^{(1)}}(G_{j,c}^{-1}(x)) \left| \det J_{G_{j,c}^{-1}}(x) \right| = p_{X^{(1)}}(x_{inv}) \left| \det J_{G_{j,c}^{-1}}(x) \right|,$$

$$\tilde{p}_{\text{do}(\text{PA}_j=c)}^{(2)}(x) = p_{X^{(2)}}(G_{j,c}^{-1}(x)) \left| \det J_{G_{j,c}^{-1}}(x) \right| = p_{X^{(2)}}(x_{inv}) \left| \det J_{G_{j,c}^{-1}}(x) \right|.$$

The ratio of the densities is $\mathcal{C}(x) = \frac{p_{X^{(2)}}(x_{inv})}{p_{X^{(1)}}(x_{inv})}$, as the Jacobian $\left| \det J_{G_{j,c}^{-1}}(x) \right|$ cancels. Now, we apply the change of variables for the CNF map $\widehat{T}^{(1)}$ at the point $x_{inv}$, using $u_{inv} = \widehat{T}^{(1)}(x_{inv})$:

$$p_{X^{(1)}}(x_{inv}) = p_{U^{(1)}}(\widehat{T}^{(1)}(x_{inv})) \left| \det J_{\widehat{T}^{(1)}}(x_{inv}) \right| = p_{U^{(1)}}(u_{inv}) \left| \det J_{\widehat{T}^{(1)}}(x_{inv}) \right|,$$

$$p_{X^{(2)}}(x_{inv}) = p_{\widehat{U}^{(2)}}(\widehat{T}^{(1)}(x_{inv})) \left| \det J_{\widehat{T}^{(1)}}(x_{inv}) \right| = p_{\widehat{U}^{(2)}}(u_{inv}) \left| \det J_{\widehat{T}^{(1)}}(x_{inv}) \right|.$$

Substituting these into the ratio for $\mathcal{C}(x)$, the second Jacobian $\left| \det J_{\widehat{T}^{(1)}}(x_{inv}) \right|$ also cancels, yielding $\mathcal{C}(x) = \frac{p_{\widehat{U}^{(2)}}(u_{inv})}{p_{U^{(1)}}(u_{inv})}$.
$\triangle$

**Forward direction ( $\implies$ ).**  Suppose a mechanism shift at $X_j$ occurs, i.e., $P(X_j^{(1)} \mid \text{PA}_j^{(1)}) \neq P(X_j^{(2)} \mid \text{PA}_j^{(2)})$. For the shifted node $j$, $P_{\widehat{U}_j^{(2)}} \neq P_{U_j^{(1)}}$. Consequently, the joint noise laws differ, $P_{\widehat{U}^{(2)}} \neq P_{U^{(1)}}$, meaning $p_{\widehat{U}^{(2)}}(u) \not\equiv p_{U^{(1)}}(u)$ on a set of positive measure. By the lemma, $\mathcal{C}(x) = \frac{p_{\widehat{U}^{(2)}}(u_{inv})}{p_{U^{(1)}}(u_{inv})}$. Since $u_{inv} = (\widehat{T}^{(1)} \circ G_{j,c}^{-1})(x)$ is a diffeomorphism, $u_{inv}$ maps sets of positive measure to sets of positive measure. Thus, $\mathcal{C}(x) \not\equiv 1$ on a set of positive measure. This implies the interventional densities differ, $\tilde{p}_{\text{do}(\text{PA}_j=c)}^{(2)} \neq p_{\text{do}(\text{PA}_j=c)}^{(1)}$, so $\mathcal{D}_{j,c} > 0$. Therefore

$$\mathbb{E}_{c \sim P(\text{PA}_j^{(1)})}[\mathcal{D}_{j,c}] \geq \int_A \mathcal{D}_{j,c} \, dP(\text{PA}_j^{(1)} = c) > 0.$$

**Converse direction ( $\impliedby$ ).**  Assume $\mathbb{E}_c[\mathcal{D}_{j,c}] > 0$. Then the set $B := \{c : \mathcal{D}_{j,c} > 0\}$ has positive $P(\text{PA}_j^{(1)})$-measure. For any $c \in B$, $\mathcal{D}_{j,c} > 0$ implies $P_{\text{do}(\text{PA}_j=c)}^{(1)} \neq \widetilde{P}_{\text{do}(\text{PA}_j=c)}^{(2)}$. By the lemma, this means $\mathcal{C}(x) = \frac{p_{\widehat{U}^{(2)}}(u_{inv})}{p_{U^{(1)}}(u_{inv})} \not\equiv 1$. Since $u_{inv}$ covers the support of the noise distributions, this implies $p_{\widehat{U}^{(2)}}(\cdot) \neq p_{U^{(1)}}(\cdot)$ as functions, and thus $P_{\widehat{U}^{(2)}} \neq P_{U^{(1)}}$. Assuming modularity ($P_{\widehat{U}_k^{(2)}} = P_{U_k^{(1)}}$ for $k \neq j$), the fact $P_{\widehat{U}^{(2)}} \neq P_{U^{(1)}}$ necessarily implies that the $j$-th noise components must differ: $P_{\widehat{U}_j^{(2)}} \neq P_{U_j^{(1)}}$. This corresponds to a mechanism shift at node $j$: $P(X_j^{(1)} \mid \text{PA}_j^{(1)}) \neq P(X_j^{(2)} \mid \text{PA}_j^{(2)})$.

### B.3. Proof of Theorem 3.2

**Theorem 3.2** For a shifted node $X_j$, the statistical independence of its inferred noise $\tilde{\varepsilon}_j^{(2)}$ and its parents $\text{PA}_j^{(2)}$ indicates the type of shift:

$$\tilde{\varepsilon}_j^{(2)} \not\perp \text{PA}_j^{(2)} \iff \text{the shift is a function shift.}$$
$$\tilde{\varepsilon}_j^{(2)} \perp \text{PA}_j^{(2)} \iff \text{the shift is a noise shift.}$$

We infer the noises in environment 2 using the SCM from environment 1: $[\tilde{\varepsilon}_1^{(2)}, ..., \tilde{\varepsilon}_J^{(2)}] = \tilde{f}^{-1}(X_1^{(2)}, ..., X_J^{(2)})$. We denote $\tilde{\varepsilon}_j^{(2)}$ as the inferred noise of environment 2 data based on $\mathcal{M}^{(1)}$.

$$X_j^{(1)} = g_j(h_j^{(1)}(\text{PA}_j^{(1)}), \varepsilon_j^{(1)})$$
$$X_j^{(2)} = g_j(h_j^{(2)}(\text{PA}_j^{(2)}), \varepsilon_j^{(2)}) = g_j(h_j^{(1)}(\text{PA}_j^{(2)}), \tilde{\varepsilon}_j^{(2)}).$$

$\varepsilon_j^{(2)}$ and $\tilde{\varepsilon}_j^{(2)}$ lie in the same probability space, while $\varepsilon_j^{(1)}$ and $\varepsilon_j^{(2)}$ lie in different probability spaces. Let $(\Omega^{(2)}, \mathcal{F}^{(2)}, \mathbb{P}^{(2)})$ be the underlying probability space of environment 2, and let

$$\varepsilon_j^{(2)} : \Omega^{(2)} \to \mathbb{R}$$

be random variables on this space such that

$$X_j^{(2)} = g_j\big(h_j^{(2)}(\mathrm{PA}_j^{(2)}), \varepsilon_j^{(2)}\big) \quad \text{a.s.}$$

By Assumption 2.2, for every fixed $p \in \mathbb{R}^{d_j}$ the map

$$\varepsilon \mapsto g_j\big(h_j^{(1)}(p), \varepsilon\big)$$

is bijective, so its inverse in the second argument exists. We therefore define the residual noise $\tilde{\varepsilon}_j^{(2)} : \Omega^{(2)} \to \mathbb{R}$ by

$$\tilde{\varepsilon}_j^{(2)}(\omega) := \big(g_j(h_j^{(1)}(\mathrm{PA}_j^{(2)}(\omega)), \cdot)\big)^{-1}\big(X_j^{(2)}(\omega)\big), \qquad \omega \in \Omega^{(2)}.$$

Then, by construction,

$$X_j^{(2)}(\omega) = g_j\big(h_j^{(1)}(\mathrm{PA}_j^{(2)}(\omega)), \tilde{\varepsilon}_j^{(2)}(\omega)\big) \quad \text{for all } \omega \in \Omega^{(2)},$$

so both $\varepsilon_j^{(2)}$ and $\tilde{\varepsilon}_j^{(2)}$ are random variables on the same probability space $(\Omega^{(2)}, \mathcal{F}^{(2)}, \mathbb{P}^{(2)})$.

UNDER FUNCTION SHIFT

We begin by defining the function shift $\Delta h_j$ as the deviation between the mechanism functions in the two environments:

$$\Delta h_j(\mathrm{PA}_j^{(2)}) := h_j^{(2)}(\mathrm{PA}_j^{(2)}) - h_j^{(1)}(\mathrm{PA}_j^{(2)}). \tag{16}$$

By equating the observed variable $X_j^{(2)}$ generated from the true mechanism with its reconstruction using the inferred noise $\tilde{\varepsilon}_j^{(2)}$ and the base mechanism, we establish the following identity:

$$g_j\big(h_j^{(1)}(\mathrm{PA}_j^{(2)}) + \Delta h_j(\mathrm{PA}_j^{(2)}), \varepsilon_j^{(2)}\big) = g_j\big(h_j^{(1)}(\mathrm{PA}_j^{(2)}), \tilde{\varepsilon}_j^{(2)}\big).$$

To rigorously analyze the dependence of the noise difference $\eta^* := \tilde{\varepsilon}_j^{(2)} - \varepsilon_j^{(2)}$ on the parents, we define the implicit function $F$ as follows:

$$F(\eta, \mathbf{pa}; \varepsilon_j^{(2)}) := g_j\big(h_j^{(1)}(\mathbf{pa}) + \Delta h_j(\mathbf{pa}), \varepsilon_j^{(2)}\big) - g_j\big(h_j^{(1)}(\mathbf{pa}), \varepsilon_j^{(2)} + \eta\big). \tag{17}$$

Note that $F(\eta^*, \mathbf{pa}; \varepsilon_j^{(2)}) = 0$. Under Assumption 2.2, since $g_j$ is differentiable and invertible with respect to the noise term, its partial derivative is non-zero:

$$\frac{\partial F}{\partial \eta}(\eta^*, \mathbf{pa}; \varepsilon_j^{(2)}) \neq 0. \tag{18}$$

Thus, by the Implicit Function Theorem, there exists a function $\Phi$ such that $\eta^* = \Phi(\mathbf{pa}, \varepsilon_j^{(2)})$. Since $\Delta h_j \notin \mathcal{T}_{g_j}$, $\Phi(\mathbf{pa}, \varepsilon_j^{(2)})$ cannot be reduced to a function of $\varepsilon_j^{(2)}$ alone independent of $\mathbf{pa}$. Crucially, $\Phi$ depends on $\mathbf{pa}$ because $F$ involves $\mathbf{pa}$ through both the original mechanism $h_j^{(1)}$ and the shift $\Delta h_j$.

Finally, since the true noise is independent of parents ($\varepsilon_j^{(2)} \perp \mathrm{PA}_j^{(2)}$) while the difference term depends on them ($\eta^* = \Phi(\mathbf{pa}, \varepsilon_j^{(2)})$), the inferred noise $\tilde{\varepsilon}_j^{(2)} = \varepsilon_j^{(2)} + \Phi(\mathrm{PA}_j^{(2)}, \varepsilon_j^{(2)})$ must be dependent on the parents. Therefore, we conclude:

$$\tilde{\varepsilon}_j^{(2)} \not\perp \mathrm{PA}_j^{(2)}.$$

For example, we assume linear functions with Gaussian noises.

$$X_j^{(1)} = \beta_j^{(1)} \mathrm{PA}_j^{(1)} + \varepsilon_j^{(1)}, \ \varepsilon_j^{(1)} \sim \mathcal{N}(0, s^2)$$
$$X_j^{(2)} = \beta_j^{(2)} \mathrm{PA}_j^{(2)} + \varepsilon_j^{(2)}, \ \varepsilon_j^{(2)} \sim \mathcal{N}(0, s^2)$$
$$= \beta_j^{(1)} \mathrm{PA}_j^{(2)} + \tilde{\varepsilon}_j^{(2)}.$$

Then, $\tilde{\varepsilon}_j^{(2)} \sim \mathcal{N}((\beta_j^{(2)} - \beta_j^{(1)}) \mathrm{PA}_j^{(2)}, s^2)$. Since $\tilde{\varepsilon}_j^{(2)}$ and $\varepsilon_j^{(2)}$ lie in the same probability space, $\eta^* := \tilde{\varepsilon}_j^{(2)} - \varepsilon_j^{(2)} = (\beta^{(2)} - \beta^{(1)}) \mathrm{PA}_j^{(2)}$.

### UNDER NOISE SHIFT

Under a pure noise shift at node $X_j$, we assume that the structural functions are unchanged across environments, $h_j^{(2)} = h_j^{(1)}$.

$$X_j^{(2)} = g_j(h_j^{(2)}(\mathrm{PA}_j^{(2)}), \varepsilon_j^{(2)}) = g_j(h_j^{(1)}(\mathrm{PA}_j^{(2)}), \varepsilon_j^{(2)}).$$

By the definition of the residual noise $\tilde{\varepsilon}_j^{(2)}$,

$$X_j^{(2)} = g_j(h_j^{(1)}(\mathrm{PA}_j^{(2)}), \tilde{\varepsilon}_j^{(2)})$$

For each fixed parent value $\mathbf{pa}$, the map $\varepsilon \mapsto g_j(h_j^{(1)}(\mathbf{pa}), \varepsilon)$ is invertible by Assumption 2.2. Thus, it concludes that $\tilde{\varepsilon}_j^{(2)} = \varepsilon_j^{(2)}$ which is independent of parent variables.

### B.4. Proof of Theorem 3.4

Our strategy is that under a function-only shift if $q_j(\tilde{\varepsilon}_j^{(2)}) = \tilde{N}_j^{(2)}$ is expressed as a linear function of $q_j(\varepsilon_j^{(2)}) = N_j^{(2)}$, then we can standardize and square it to get a chi-square distribution. However, when a function and noise simultaneously shift, then this process doesn't ensure chi-square distribution since it induces an extra randomness. We let $b_{j,0}(\mathbf{pa})$ and $b_{j,1}(\mathbf{pa})$ are arbitrary functions of $\mathbf{pa}$. Then, under a function-only shift, we assume $\tilde{N}_j^{(2)}$ is expressed as a following linear expression:

$$\tilde{N}_j^{(2)} = N_j^{(2)} b_{j,1}(\mathbf{pa}) + b_{j,0}(\mathbf{pa}).$$

Since $N_j^{(2)} \perp \mathrm{PA}_j^{(2)}$ and $E[N_j^{(2)}] = 0$,

$$E[\tilde{N}_j^{(2)}|\mathrm{PA}_j^{(2)}] = E[N_j^{(2)} b_{j,1}(\mathrm{PA}_j^{(2)}) + b_{j,0}(\mathrm{PA}_j^{(2)})|\mathrm{PA}_j^{(2)}] = b_{j,0}(\mathrm{PA}_j^{(2)}).$$

Also, since $\mathrm{Var}(N_j^{(2)}) = 1$,

$$\mathrm{Var}(\tilde{N}_j^{(2)}|\mathrm{PA}_j^{(2)}) = (b_{j,1}(\mathrm{PA}_j^{(2)}))^2 \mathrm{Var}(N_j^{(2)}) = (b_{j,1}(\mathrm{PA}_j^{(2)}))^2.$$

Now, consider the test statistics $\tilde{q}_j(\tilde{\varepsilon}_j^{(2)})$ defined as follows:

$$\tilde{q}_j(\tilde{\varepsilon}_j^{(2)}) := \frac{\tilde{N}_j^{(2)} - E[\tilde{N}_j^{(2)}|\mathrm{PA}_j^{(2)}]}{\sqrt{\mathrm{Var}(\tilde{N}_j^{(2)}|\mathrm{PA}_j^{(2)})}} = \frac{b_{j,1}(\mathrm{PA}_j^{(2)})}{|b_{j,1}(\mathrm{PA}_j^{(2)})|} N_j^{(2)} = \mathrm{sgn}(b_{j,1}(\mathrm{PA}_j^{(2)})) N_j^{(2)}.$$

To handle the sign function, we square the standardized noise.

$$(\tilde{q}_j(\tilde{\varepsilon}_j^{(2)}))^2 = (\mathrm{sgn}(b_{j,1}(\mathrm{PA}_j^{(2)})) N_j^{(2)})^2 = (N_j^{(2)})^2.$$

Assumption 3.3 describes

$$g_j^*(h_j, N_j) = \Psi_j(\beta_{j,1}(h_j) \cdot N_j + \beta_{j,0}(h_j)).$$

When we consider a function-only shift in environment 2,

$$
\begin{aligned}
X_j^{(2)} = g_j^*(h_j^{(2)}, N_j^{(2)}) &= \Psi_j(\beta_{j,1}(h_j^{(2)}) \cdot N_j^{(2)} + \beta_{j,0}(h_j^{(2)})) \\
&= g_j^*(h_j^{(1)}, \tilde{N}_j^{(2)}) = \Psi_j(\beta_{j,1}(h_j^{(1)}) \cdot \tilde{N}_j^{(2)} + \beta_{j,0}(h_j^{(1)})).
\end{aligned}
$$

Since $\Psi_j$ is invertible,

$$\beta_{j,1}(h_j^{(2)}) \cdot N_j^{(2)} + \beta_{j,0}(h_j^{(2)}) = \beta_{j,1}(h_j^{(1)}) \cdot \tilde{N}_j^{(2)} + \beta_{j,0}(h_j^{(1)}).$$

Then,

$$\tilde{N}_j^{(2)} = N_j^{(2)} b_{j,1}(\mathbf{pa}_j) + b_{j,0}(\mathbf{pa}_j).$$

The converse does not hold in general. If the shift in $N_j^{(2)}$ is limited to location and scale, the square of the standardized variable $(\tilde{N}_j^{(2)})$ follows $\chi^2(1)$. However, in the case of a function-noise shift affecting higher order moments (i.e., third moment or higher), the standardized distribution deviates from normality. Consequently, it implies

$$(\tilde{q}_j(\tilde{\varepsilon}_j^{(2)}))^2 \not\sim \chi^2(1).$$

## B.5. Proof of Proposition 3.7 and Corollary 3.8

In this appendix, we provide a proof of Proposition 3.7 and Corollary 3.8.

### B.5.1. SETUP AND NOTATION

Fix a node $j$ and an environment $k \in \{1, 2\}$. We write $N := N_j^{(k)}$ for the true residual noise, $\hat{N} := \hat{N}_j^{(k)} = \hat{q}_j(\tilde{\varepsilon}_j^{(k)})$ for its CNF-based estimate, and $Z := Z^{(k)} = \mathrm{PA}_j^{(k)}$ for the parents. Let $(N', \hat{N}', Z')$ and $(N'', \hat{N}'', Z'')$ denote independent copies sharing the joint distribution of $(N, \hat{N}, Z)$.

The squared distance covariance of random vectors $U, V$ admits the expectation form (Székely et al., 2007)

$$
\begin{aligned}
\mathcal{V}^2(U, V) = {}& \mathbb{E}[\|U - U'\| \cdot \|V - V'\|] + \mathbb{E}[\|U - U'\|] \cdot \mathbb{E}[\|V - V'\|] \\
& - 2\,\mathbb{E}[\|U - U'\| \cdot \|V - V''\|],
\end{aligned}
\tag{19}
$$

and the distance correlation is defined as

$$
\mathcal{R}(U, V) := \begin{cases} \sqrt{\dfrac{\mathcal{V}^2(U, V)}{\sqrt{\mathcal{V}^2(U, U) \cdot \mathcal{V}^2(V, V)}}} & \text{if } \mathcal{V}^2(U, U) \cdot \mathcal{V}^2(V, V) > 0, \\ 0 & \text{otherwise.} \end{cases}
\tag{20}
$$

Under finite second moments, $0 \le \mathcal{R}(U, V) \le 1$, with equality to zero if and only if $U$ and $V$ are independent.

**Population-level analysis.** Throughout this appendix, $\mathcal{V}^2(U, V)$ and $\mathcal{R}(U, V)$ denote the *population* distance covariance and distance correlation, defined with respect to the joint distribution of $(U, V)$. Our goal is to quantify how the CNF approximation error $\delta_n$ propagates to the test statistic at the population level. In practice, these quantities are replaced by their empirical V-statistic estimators computed from finite test samples, which introduces an additional $O_p(1/\sqrt{m})$ sampling error by standard results (Székely et al., 2007). This sampling error is orthogonal to the CNF-induced error analyzed here and does not affect the leading-order behavior as both $n, m \to \infty$.

Moreover, all $O_p$ and $o_p$ statements below are with respect to the randomness of CNF training (i.e., the sample used to estimate $\hat{q}_j$), which is captured by the rate $\delta_n$ in Assumption 3.5.

The analysis relies on two conditions. Assumption 3.5 ($L^2$ CNF consistency) is interpreted here as follows: there exists a deterministic sequence $\delta_n \to 0$ as the CNF training sample size $n \to \infty$ such that, with probability one over CNF training,

$$\mathbb{E}_{X^{(k)}}\big[(\hat{N}_j^{(k)} - N_j^{(k)})^2 \,\big|\, \hat{q}_j\big] \le \delta_n^2, \tag{21}$$

where the conditional expectation is taken over the data distribution of environment $k$ given the trained CNF map $\hat{q}_j$. Assumption 3.6 (finite second moments) ensures that $\mathbb{E}[(N_j^{(k)})^2] < \infty$ and $\mathbb{E}[\|Z^{(k)}\|^2] < \infty$ for each $j$ and $k$.

Throughout, define $C_Z := \sqrt{\mathbb{E}[\|Z - Z'\|^2]}$ and $C_N := \sqrt{\mathbb{E}[(N - N')^2]}$, both finite by Assumption 3.6. All bounds in Lemmas B.1 and B.2 hold conditionally on a CNF realization satisfying Assumption 3.5; by the $L^2$ consistency, such realizations occur with probability one in the limit.

### B.5.2. PROOF

**Lemma B.1** (Distance Covariance Error Bound). *Under Assumptions 3.5 and 3.6,*

$$\big|\mathcal{V}^2(\hat{N}, Z) - \mathcal{V}^2(N, Z)\big| \le 8\,C_Z\,\delta_n. \tag{22}$$

*Proof.* By the triangle inequality applied to the three terms in the definition (19),

$$\big|\mathcal{V}^2(\hat{N}, Z) - \mathcal{V}^2(N, Z)\big| \le |A_1| + |A_2| + 2|A_3|, \tag{23}$$

where

$$A_1 := \mathbb{E}[|\hat{N} - \hat{N}'|\|Z - Z'\|] - \mathbb{E}[|N - N'|\|Z - Z'\|], \tag{24}$$

$$A_2 := \big(\mathbb{E}[|\hat{N} - \hat{N}'|] - \mathbb{E}[|N - N'|]\big) \cdot \mathbb{E}[\|Z - Z'\|], \tag{25}$$

$$A_3 := \mathbb{E}[|\hat{N} - \hat{N}'|\|Z - Z''\|] - \mathbb{E}[|N - N'|\|Z - Z''\|]. \tag{26}$$

*Bound on $|A_1|$.* By the reverse triangle inequality $\big||\hat{N} - \hat{N}'| - |N - N'|\big| \le |\hat{N} - N| + |\hat{N}' - N'|$,

$$|A_1| \le \mathbb{E}[|\hat{N} - N|\|Z - Z'\|] + \mathbb{E}[|\hat{N}' - N'|\|Z - Z'\|]. \tag{27}$$

Applying Cauchy–Schwarz to each term,

$$\mathbb{E}[|\hat{N} - N|\|Z - Z'\|] \le \sqrt{\mathbb{E}[(\hat{N} - N)^2]} \cdot \sqrt{\mathbb{E}[\|Z - Z'\|^2]} \le \delta_n C_Z, \tag{28}$$

where the last inequality invokes Assumption 3.5 and the definition of $C_Z$. By the identical distribution of $(\hat{N}, N)$ and $(\hat{N}', N')$, the second term has the same bound. Therefore

$$|A_1| \le 2\delta_n C_Z. \tag{29}$$

*Bound on $|A_2|$.* Since $\mathbb{E}[\|Z - Z'\|] \le \sqrt{\mathbb{E}[\|Z - Z'\|^2]} = C_Z$ by Jensen's inequality,

$$\begin{aligned}
|A_2| &\le \big|\mathbb{E}[|\hat{N} - \hat{N}'|] - \mathbb{E}[|N - N'|]\big| \cdot C_Z \\
&\le \mathbb{E}\big[\big||\hat{N} - \hat{N}'| - |N - N'|\big|\big] \cdot C_Z \\
&\le \big(\mathbb{E}[|\hat{N} - N|] + \mathbb{E}[|\hat{N}' - N'|]\big) \cdot C_Z \\
&\le 2\sqrt{\mathbb{E}[(\hat{N} - N)^2]} \cdot C_Z \le 2\delta_n C_Z,
\end{aligned} \tag{30}$$

where the third inequality is the reverse triangle inequality and the penultimate inequality is Jensen's applied to $|\cdot|$.

*Bound on $|A_3|$.* Replacing $Z'$ with $Z''$ in the argument for $A_1$, and using the identical joint distribution of $(Z, Z'')$ and $(Z, Z')$ (since $Z''$ is another i.i.d. copy),

$$|A_3| \leq 2\delta_n C_Z. \tag{31}$$

Combining (29)–(31),

$$\left|\mathcal{V}^2(\hat{N}, Z) - \mathcal{V}^2(N, Z)\right| \leq 2\delta_n C_Z + 2\delta_n C_Z + 2 \cdot 2\delta_n C_Z = 8\delta_n C_Z. \tag{32}$$

$\square$

**Lemma B.2** (Distance Variance Error Bound). *Under Assumptions 3.5 and 3.6,*

$$\left|\mathcal{V}^2(\hat{N}, \hat{N}) - \mathcal{V}^2(N, N)\right| \leq 16\,C_N\,\delta_n + o(\delta_n). \tag{33}$$

*Proof.* Setting $V = U$ in (19) and applying the triangle inequality,

$$\left|\mathcal{V}^2(\hat{N}, \hat{N}) - \mathcal{V}^2(N, N)\right| \leq |B_1| + |B_2| + 2|B_3|, \tag{34}$$

where

$$B_1 := \mathbb{E}[|\hat{N} - \hat{N}'|^2] - \mathbb{E}[|N - N'|^2], \tag{35}$$

$$B_2 := \left(\mathbb{E}[|\hat{N} - \hat{N}'|]\right)^2 - \left(\mathbb{E}[|N - N'|]\right)^2, \tag{36}$$

$$B_3 := \mathbb{E}[|\hat{N} - \hat{N}'| \cdot |\hat{N} - \hat{N}''|] - \mathbb{E}[|N - N'| \cdot |N - N''|]. \tag{37}$$

The key observation is that each term now involves two instances of $\hat{N}$ (versus two of $N$), so the reverse triangle inequality accumulates twice as many error contributions compared to Lemma B.1.

*Bound on $|B_1|$.* Factoring the difference of squares,

$$\begin{aligned}
|B_1| &\leq \mathbb{E}\big[(|\hat{N} - \hat{N}'| + |N - N'|) \cdot \big||\hat{N} - \hat{N}'| - |N - N'|\big|\big] \\
&\leq \mathbb{E}\big[(|\hat{N} - \hat{N}'| + |N - N'|)(|\hat{N} - N| + |\hat{N}' - N'|)\big].
\end{aligned} \tag{38}$$

By Cauchy–Schwarz,

$$\begin{aligned}
|B_1| &\leq \sqrt{\mathbb{E}[(|\hat{N} - \hat{N}'| + |N - N'|)^2]} \cdot \sqrt{\mathbb{E}[(|\hat{N} - N| + |\hat{N}' - N'|)^2]} \\
&\leq \sqrt{2\,\mathbb{E}[(\hat{N} - \hat{N}')^2] + 2\,\mathbb{E}[(N - N')^2]} \cdot \sqrt{4\delta_n^2} \\
&\leq 2\delta_n \cdot (2C_N + o(1)) = 4\delta_n C_N + o(\delta_n),
\end{aligned} \tag{39}$$

where the final step uses $\mathbb{E}[(\hat{N} - \hat{N}')^2] \to \mathbb{E}[(N - N')^2] = C_N^2$ as $n \to \infty$, a direct consequence of Assumption 3.5.

*Bound on $|B_2|$.* Using $a^2 - b^2 = (a+b)(a-b)$,

$$\begin{aligned}
|B_2| &= \big|\mathbb{E}[|\hat{N} - \hat{N}'|] + \mathbb{E}[|N - N'|]\big| \cdot \big|\mathbb{E}[|\hat{N} - \hat{N}'|] - \mathbb{E}[|N - N'|]\big| \\
&\leq (2C_N + o(1)) \cdot 2\delta_n = 4\delta_n C_N + o(\delta_n),
\end{aligned} \tag{40}$$

where the second inequality applies the same reverse-triangle-plus-Jensen argument as in (30) to the second factor, and Jensen's inequality ($\mathbb{E}[|\cdot|] \leq \sqrt{\mathbb{E}[|\cdot|^2]}$) together with Assumption 3.5 to the first factor.

*Bound on $|B_3|$.* By the reverse triangle inequality applied to the product $|\hat{N} - \hat{N}'| \cdot |\hat{N} - \hat{N}''|$,

$$\begin{aligned}
|B_3| &\leq \mathbb{E}\big[|\hat{N} - N|(|\hat{N} - \hat{N}''| + |N - N''|)\big] + \mathbb{E}\big[|\hat{N}' - N'| \cdot |\hat{N} - \hat{N}''|\big] \\
&\quad + \mathbb{E}\big[|N - N'| \cdot |\hat{N} - N|\big] + \mathbb{E}\big[|N - N'| \cdot |\hat{N}'' - N''|\big].
\end{aligned} \tag{41}$$

Each of the four terms is bounded by Cauchy–Schwarz and Assumption 3.5, giving $|B_3| \leq 4\delta_n C_N + o(\delta_n)$.

Summing the three contributions,

$$|B_1| + |B_2| + 2|B_3| \leq 4\delta_n C_N + 4\delta_n C_N + 2 \cdot 4\delta_n C_N + o(\delta_n) = 16\delta_n C_N + o(\delta_n). \tag{42}$$

$\square$

Lemma B.2 implies in particular that $\mathcal{V}^2(\hat{N}, \hat{N}) \xrightarrow{p} \mathcal{V}^2(N, N)$ as $n \to \infty$.

**Lemma B.3** (Denominator Stability). *Under Assumptions 3.5 and 3.6, and assuming $\mathcal{V}^2(N, N) > 0$ and $\mathcal{V}^2(Z, Z) > 0$, define*

$$D(U, V) := \sqrt{\mathcal{V}^2(U, U) \cdot \mathcal{V}^2(V, V)}. \tag{43}$$

*Then $D(\hat{N}, Z) \xrightarrow{p} D(N, Z) > 0$ as $n \to \infty$, and moreover $|D(\hat{N}, Z) - D(N, Z)| = O_p(\delta_n)$.*

*Proof.* By Lemma B.2, $\mathcal{V}^2(\hat{N}, \hat{N}) = \mathcal{V}^2(N, N) + O_p(\delta_n)$, and $\mathcal{V}^2(Z, Z)$ is deterministic and positive. Applying the mean value theorem to $u \mapsto \sqrt{u}$ on a neighborhood of $\mathcal{V}^2(N, N) > 0$ yields $|\sqrt{\mathcal{V}^2(\hat{N}, \hat{N})} - \sqrt{\mathcal{V}^2(N, N)}| = O_p(\delta_n)$. Multiplying by the constant $\sqrt{\mathcal{V}^2(Z, Z)}$ gives the stated rate. $\square$

The non-degeneracy conditions $\mathcal{V}^2(N, N), \mathcal{V}^2(Z, Z) > 0$ hold whenever $N$ and $Z$ are non-trivial (not almost surely constant), which is the setting of interest for dissection.

**Lemma B.4** (Rate of Distance Correlation Convergence). *Under Assumptions 3.5 and 3.6, and assuming non-degeneracy of $N$ and $Z$,*

$$\left| \mathcal{R}(\hat{N}, Z) - \mathcal{R}(N, Z) \right| = O_p(\sqrt{\delta_n}). \tag{44}$$

*Moreover, if $\mathcal{R}(N, Z) > 0$, the rate sharpens to $O_p(\delta_n)$.*

*Proof.* Let $r := \mathcal{V}^2(N, Z)/D(N, Z)$ and $\hat{r} := \mathcal{V}^2(\hat{N}, Z)/D(\hat{N}, Z)$, so that $\mathcal{R}(N, Z) = \sqrt{r}$ and $\mathcal{R}(\hat{N}, Z) = \sqrt{\hat{r}}$.

We first bound $|\hat{r} - r|$. Writing

$$\hat{r} - r = \frac{\mathcal{V}^2(\hat{N}, Z) - \mathcal{V}^2(N, Z)}{D(\hat{N}, Z)} + \mathcal{V}^2(N, Z) \cdot \frac{D(N, Z) - D(\hat{N}, Z)}{D(\hat{N}, Z) \, D(N, Z)}, \tag{45}$$

and applying Lemmas B.1 and B.3,

$$|\hat{r} - r| = O_p(\delta_n), \tag{46}$$

since both numerators are $O_p(\delta_n)$ and the denominators converge in probability to positive constants.

*Case 1: $r = 0$ ($\mathcal{R}(N, Z) = 0$).* Since $\hat{r} \geq 0$ almost surely ($\mathcal{V}^2 \geq 0$) and $\hat{r} = |\hat{r} - r| = O_p(\delta_n)$ by (46),

$$\mathcal{R}(\hat{N}, Z) = \sqrt{\hat{r}} = O_p(\sqrt{\delta_n}). \tag{47}$$

*Case 2: $r > 0$ ($\mathcal{R}(N, Z) > 0$).* Using the identity $\sqrt{\hat{r}} - \sqrt{r} = \frac{\hat{r} - r}{\sqrt{\hat{r}} + \sqrt{r}}$ and noting that $\sqrt{\hat{r}} + \sqrt{r} \xrightarrow{p} 2\sqrt{r} > 0$, we obtain

$$\left| \sqrt{\hat{r}} - \sqrt{r} \right| = \frac{|\hat{r} - r|}{\sqrt{\hat{r}} + \sqrt{r}} = O_p(\delta_n). \tag{48}$$

*Uniform bound.* Since $\delta_n \leq \sqrt{\delta_n}$ for $\delta_n \leq 1$ (which holds for $n$ sufficiently large as $\delta_n \to 0$), the rate $O_p(\sqrt{\delta_n})$ subsumes both cases, establishing (44). $\square$

*Proof of Proposition 3.7.* Applying Lemma B.4 separately to environments $k = 1$ and $k = 2$,

$$\hat{\mathcal{R}}_1 := \mathcal{R}(\hat{N}^{(1)}, Z^{(1)}) \xrightarrow{p} \mathcal{R}(N^{(1)}, Z^{(1)}) =: \mathcal{R}_1, \tag{49}$$

$$\hat{\mathcal{R}}_2 := \mathcal{R}(\hat{N}^{(2)}, Z^{(2)}) \xrightarrow{p} \mathcal{R}(N^{(2)}, Z^{(2)}) =: \mathcal{R}_2. \tag{50}$$

Componentwise convergence in probability implies joint convergence, so by the continuous mapping theorem applied to $(a, b) \mapsto |a - b|$,

$$\Delta\hat{\mathcal{R}} = |\hat{\mathcal{R}}_2 - \hat{\mathcal{R}}_1| \xrightarrow{p} |\mathcal{R}_2 - \mathcal{R}_1| = \Delta\mathcal{R}^*. \tag{51}$$

We further characterize the rate of convergence:

- **Pure noise shift** ($\mathcal{R}_1 = \mathcal{R}_2 = 0$, so $\Delta\mathcal{R}^* = 0$): Case 1 of Lemma B.4 applies to both environments, giving $\Delta\hat{\mathcal{R}} = O_p(\sqrt{\delta_n})$.

- **Function shift** ($\mathcal{R}_2 > 0$, so $\Delta\mathcal{R}^* > 0$): Case 2 applies to environment 2, so $|\hat{\mathcal{R}}_2 - \mathcal{R}_2| = O_p(\delta_n)$, while $\hat{\mathcal{R}}_1 = O_p(\sqrt{\delta_n})$ from Case 1 (since under the null we have $\mathcal{R}_1 = 0$). Combining via the triangle inequality yields $|\Delta\hat{\mathcal{R}} - \Delta\mathcal{R}^*| = O_p(\sqrt{\delta_n})$.

In both regimes, $\Delta\hat{\mathcal{R}} \xrightarrow{p} \Delta\mathcal{R}^*$ as $n \to \infty$. $\qquad\square$

*Proof of Corollary 3.8.* We verify the two claims for any fixed rejection threshold $\tau > 0$.

*Type I error (pure noise shift, $\Delta\mathcal{R}^* = 0$).* By Proposition 3.7, $\Delta\hat{\mathcal{R}} \xrightarrow{p} 0$. Hence

$$\Pr(\Delta\hat{\mathcal{R}} > \tau) = \Pr(|\Delta\hat{\mathcal{R}} - 0| > \tau) \to 0 \quad \text{as } n \to \infty. \tag{52}$$

*Type II error (function shift, $\Delta\mathcal{R}^* > 0$).* Fix $\tau \in (0, \Delta\mathcal{R}^*)$ and let $\epsilon := \Delta\mathcal{R}^* - \tau > 0$. Since

$$\{\Delta\hat{\mathcal{R}} \leq \tau\} = \{\Delta\mathcal{R}^* - \Delta\hat{\mathcal{R}} \geq \epsilon\} \subseteq \{|\Delta\hat{\mathcal{R}} - \Delta\mathcal{R}^*| \geq \epsilon\}, \tag{53}$$

we have

$$\Pr(\Delta\hat{\mathcal{R}} \leq \tau) \leq \Pr(|\Delta\hat{\mathcal{R}} - \Delta\mathcal{R}^*| \geq \epsilon) \to 0 \quad \text{as } n \to \infty, \tag{54}$$

by Proposition 3.7. Equivalently, $\Pr(\Delta\hat{\mathcal{R}} > \tau) \to 1$. $\qquad\square$

## C. Experiment Details

For all synthetic experiments, we generate 50,000 samples per environment. Of these, 40,000 are used to train the CNF, and the remaining 10,000 are held out as a validation set used strictly to monitor the learning curve and detect overfitting.

We denote $S^{\text{method}}$ as a set of shift detected nodes by a method, and $S^{\text{method,func}}$ is a set of function shift detected nodes by a method and $S^{\text{method,noise}}$ is a set of noise shift detected nodes. This section explains how these sets are derived from each methodology in detail. Furthermore, $S^{\text{true}}$ is a true set of shifted nodes, which is identical to $S$.

### C.1. Synthetic dataset for detection and dissection

C.1.1. FANS

We implement FANS on top of the Causal Normalizing Flow (CNF) architecture of Javaloy et al. (2023). We use the following default configuration across all synthetic experiments: Neural Autoregressive Flow as the flow layer, hidden layer dimensions of $[32, 32, 32]$, batch size of 1024, learning rate of 0.01 with decay, and 3000 training epochs. The Adam optimizer is used throughout. A sensitivity analysis of these hyperparameters is provided in Appendix D.1.1.

For detection of shift, we use the Jensen-Shannon Divergence (JSD). We approximate the overall distributional shift by evaluating the JSD at three specific conditioning points $c \in \{c^1, c^2, c^3\}$. These points are selected as the 0.25, 0.5, and 0.75 quantiles of the empirical distribution of $\text{PA}_j$ in environment 1. Specifically, for a multi dimensional parent vector $\text{PA}_j$, these quantile points are computed element-wise. Let $\mathcal{D}_{j,c}$ be the JSD between the conditional distributions at a specific point $c$:

$$\mathcal{D}_{j,c} := \text{JSD}(P^{(1)}_{\text{do}(\text{PA}_j=c)}, \tilde{P}^{(2)}_{\text{do}(\text{PA}_j=c)}).$$

This indicates shifts occur in some nodes. We then define an aggregate divergence measure for node $j$, $\bar{\mathcal{D}}^{\text{FANS}}_j$, as the average JSD over these three selected points. To identify shifted nodes, we employed an elbow point method. We sort the divergence values in descending order and determine an elbow point $e$. Let $\mathcal{D}_e$ be the JSD value at this elbow point. We classify the nodes with divergence greater than $\mathcal{D}_e$ as shifted:

$$S^{\text{FANS}} := \{j : \bar{\mathcal{D}}^{\text{FANS}}_j > \mathcal{D}_e\}.$$

Furthermore, for the node index $j \in S^{\text{FANS}}$ identified as shifted, we perform a dissection to determine whether the shift is primarily due to a function shift or a noise shift. For each node index $j$ in $S^{\text{true}}$, we derive the inferred noise $q(\tilde{\varepsilon}_j^{(k)})$ for both environments $k \in \{1, 2\}$. Let $\mathcal{R}(\cdot, \cdot)$ denote the empirical distance correlation. We compute the absolute difference between the distance correlation in Environment 2 and the baseline correlation in Environment 1. If this absolute difference exceeds a predefined threshold $\tau_{\text{dis}}$ (e.g., 0.05), we determine that the independence condition is violated. Consequently, $j$ is classified as a function shift; otherwise, it is classified as a noise shift.

$$\Delta\mathcal{R}_j := \left| \mathcal{R}\left( q(\tilde{\varepsilon}_j^{(2)}), \text{PA}_j^{(2)} \right) - \mathcal{R}\left( q(\tilde{\varepsilon}_j^{(1)}), \text{PA}_j^{(1)} \right) \right|$$
$$S^{\text{FANS,func}} := \{ j \in S^{\text{true}} : \Delta\mathcal{R}_j > \tau_{\text{dis}} \}$$
$$S^{\text{FANS,noise}} := S^{\text{true}} \setminus S^{\text{FANS,func}}.$$

Alternatively, other methods such as permutation tests based on distance covariance (Ramos-Carreño & Torrecilla, 2023) can be applied.

### C.1.2. ISCAN

For the initial shift detection task, iSCAN returns a score $s_j^{\text{iSCAN}}$ for each node j, which serves as our initial test statistic for shift detection. A higher score indicates a higher likelihood of a shift. The null hypothesis $H_{0,j}^{\text{iSCAN}} : s_j^{\text{iSCAN}} < t$ (node $j$ is not shifted) is rejected if $s_j \geq t$. The threshold $t$ was derived by elbow strategy in the original iSCAN paper (Chen et al., 2023). The set of nodes identified as shifted by iSCAN is

$$S^{\text{iSCAN}} := \{ j : s_j^{\text{iSCAN}} \geq t \}.$$

For the dissection task, concerning nodes $j \in S^{\text{iSCAN}}$, we implement Algorithm 5 in (Chen et al., 2023) to detect functionally shifted edges. We model the functional relationship as nonlinear function between a node $X_j$ and its parents $\text{PA}_j$ which has a following relationship.

$$X_j^{(k)} = f_j^{(k)}(\text{PA}_j^{(k)}) + \varepsilon_j^{(k)}, \ k \in \{1, 2\}.$$

Following the iSCAN methodology, $f_j^{(k)} : \mathbb{R}^{d_j} \mapsto \mathbb{R}$ is assumed to have an additive form $f_j^{(k)}(\text{PA}_j^{(k)}) = \sum_{p=1}^{d_j} f_{j,p}^{(k)}([\text{PA}_j^{(k)}]_p)$, where $f_{j,p}^{(k)} : \mathbb{R} \mapsto \mathbb{R}$ are univariate functions. We approximate each $f_{j,p}^{(k)}$ using a B-spline expansion. Specifically, we use cubic B-spline with 4 degrees of freedom for each parent component $[\text{PA}_j^{(k)}]_p$. The $i$-th basis function is denoted as $B_{j,i,p}(x_{j,p}^{(k)})$ for a node $j$ and $p$-th parent variable. We denote $\mathbf{B}_{j,p}^{(k)} = [B_{j,1,p}(x_{j,p}^{(k)}), ..., B_{j,4,p}(x_{j,p}^{(k)})]^T$ be the column vector of these basis functions. Similarly, $\tilde{\beta}_{j,p} = [\tilde{\beta}_{j,1,p}^{(k)}, ..., \tilde{\beta}_{j,4,p}^{(k)}]^T$ be the column vector of corresponding coefficients for $p \in \{1, ..., d_j\}$. The approximation is given by:

$$f_j^{(k)}(\text{PA}_j^{(k)}) \approx \tilde{\beta}_{j,0}^{(k)} + \sum_{p=1}^{d_j} (\tilde{\beta}_{j,p}^{(k)})^T \mathbf{B}_{j,p}^{(k)}([\text{PA}_j^{(k)}]_p),$$

where $\tilde{\beta}_{j,0}^{(k)}$ is the intercept. Since our setting assumes a given DAG, we use the known parent set without estimating them. Let $\tilde{\beta}_j^{(k)}$ denote the vector concatenating all coefficient vectors $\beta_{j,p}^{(k)}$ (for $p = 1, ..., d_j$).

A function shift is characterized by a change in these estimated B-spline coefficients. The null hypothesis of no functional shift for node $j \in S^{\text{iSCAN}}$ is:

$$H_{0,j}^{\text{iSCAN,func}} : \tilde{\beta}_j^{(1)} = \tilde{\beta}_j^{(2)}.$$

This hypothesis is tested using Hotelling's T-squared test with a significance level of 0.05. A rejection of $H_{0,j}^{\text{iSCAN,func}}$ indicates a functional shift. The set of functionally shifted nodes is:

$$S^{\text{iSCAN,func}} := \{ j \in S^{\text{true}} : H_{0,j}^{\text{iSCAN, func}} \text{ is rejected at the 0.05 significance level} \}.$$

If a node $j \in S^{\text{iSCAN}}$ is identified as shifted by the initial scored-based criterion but is not found to have a functional shift, it is then classified as a noise shift. Thus, the set of noise-shifted nodes is:

$$S^{\text{iSCAN,noise}} := S^{\text{true}} \backslash S^{\text{iSCAN,func}}.$$

### C.1.3. LINEARCCP

LinearCCP (Huang et al., 2024) was originally designed for scenarios where information about parent sets is unavailable. However, as our experimental setting assumes a given DAG, we adapt LCCP by utilizing the known parent sets for each node $j$ as independent variables in a linear causal model. For each environment $k \in \{1, 2\}$, the model for node $j$ is:

$$X_j^{(k)} = \sum_{p=1}^{d_j} \beta_{j,p}^{(k)} [\text{PA}_j^{(k)}]_p + \varepsilon_j^{(k)},$$

where $\beta_{j,p}^{(k)}$ are the corresponding linear coefficients. Let $\beta_j^{(k)} = [\beta_{j,1}^{(k)}, ..., \beta_{j,d_j}^{(k)}]^T$ be the vector of these coefficients. These coefficients $\hat{\beta}_j^{(k)}$ are estimated using Ordinary Least Squares (OLS), and the corresponding residuals are denoted as $\hat{\varepsilon}_j^{(k)}$. LCCP is used to detect two types of shifts: functional shifts (changes in coefficients) and noise shifts (change in the error distribution).

The two null hypotheses of no functional shift or no noise shift for node index $j$ are:

$$H_{0,j}^{\text{LCCP,func}} : \beta_j^{(1)} = \beta_j^{(2)}$$
$$H_{0,j}^{\text{LCCP,noise}} : F_{\varepsilon_j^{(1)}}(x) = F_{\varepsilon_j^{(2)}}(x) \text{ for all } x.$$

$H_{0,j}^{\text{LCCP,func}}$ is tested using the Chow test (Chow, 1960). If the p-value from this test is less than the significance level of 0.05, the null hypothesis is rejected, indicating a functional shift. Also, $H_{0,j}^{\text{LCCP,noise}}$ is tested using a two-sample Kolmogorov-Smirnov (KS) test on the estimated residuals $\hat{\varepsilon}_j^{(1)}$ and $\hat{\varepsilon}_j^{(2)}$ at a 0.05 significance level. If any of these hypotheses is rejected, we classify the node index $j$ as being shifted.

A dissection task is executed following other methods like iSCAN and FANS. If $H_{0,j}^{\text{LCCP,func}}$ is rejected, we decide $j$ is a function shift detected node. Noise shifted nodes are then determined by difference of $S^{\text{true}}$ and $S^{\text{LCCP,func}}$

$$S^{\text{LCCP,func}} := \{j \in S^{\text{true}} | (H_{0,j}^{\text{LCCP,func}} \text{ is rejected})\}$$
$$S^{\text{LCCP,noise}} := S^{\text{true}} \backslash S^{\text{LCCP,func}}.$$

### C.1.4. SPLITKCI

We implement the Split Kernel Conditional Independence (SplitKCI) test (Pogodin et al., 2024) to detect shifted nodes between two environments. The test evaluates the null hypothesis:

$$H_{0,j}^{\text{SplitKCI}} : X_j \perp Z \mid \text{PA}_j,$$

for each node $j$, where $Z \in \{1, 2\}$ denotes the environment indicator and $\text{PA}_j$ are all the parent variables of node $j$ in the known DAG structure. The SplitKCI test is a kernel-based conditional independence test with bias reduction via data splitting.

Although SplitKCI is formulated as a conditional independence test of the form $X_j \perp Z \mid \text{PA}_j$, it effectively functions as a *two-sample conditional test*. Specifically, it tests whether the conditional distribution $P(X_j \mid \text{PA}_j)$ remains invariant across two different environments. This is achieved by embedding samples from both environments into reproducing kernel Hilbert spaces (RKHS), and measuring the dependence between the target variable $X_j$ and the environment label $Z$, conditioning on $\text{PA}_j$. Thus, SplitKCI operates the two-sample conditional hypothesis by testing for violations of node shifts in a nonparametric manner.

Specifically, we evaluate a conditional independence statistic using the Hilbert-Schmidt norm of a cross-covariance operator:

$$T_j^{\text{SplitKCI}} := \left\langle C_j^{(1)}, C_j^{(2)} \right\rangle_{\text{HS}},$$

where $C_j^{(1)}$ and $C_j^{(2)}$ are Hilbert-Schmidt operators constructed from two independent splits of the data. These operators are computed using conditional mean embeddings (CME) estimated via kernel ridge regression from $\text{PA}_j$ to $X_j$, and from $\text{PA}_j$ to $Z$. To control for bias in finite samples, the CME is estimated on two disjoint subsets of the data. This reduces dependence between estimators and enhances validity of the test. We use Gaussian kernels for both $X_j$ and $\text{PA}_j$. The significance of each $T_j^{\text{SplitKCI}}$ is assessed via the wild bootstrap method. We draw Rademacher weights $w_i$ and compute the test statistic under bootstrapped samples to form a null distribution. The p-value for each node is estimated as:

$$p_j = \frac{1}{B} \sum_{b=1}^{B} \mathbb{I}\left( T_{j,b}^{\text{boot}} \geq T_j^{\text{SplitKCI}} \right),$$

where $B$ is the number of bootstrap replicates. The node $j$ is declared shifted if $p_j < \alpha$ for a significance level $\alpha = 0.05$. The set of shifted nodes is defined as:

$$S^{\text{SplitKCI}} := \{j : p_j < 0.05\}.$$

### C.1.5. PREDITER

We evaluate the PreDITEr algorithm proposed by Varici et al. (2022), which detects shifted nodes between two environments. PreDITEr exploits the fact that soft interventions induce sparse changes in the precision matrix of a linear SEM, thereby formulating the task as a *precision difference estimation* problem.

Specifically, PreDITEr models the observed precision matrix in each environment $k$ as $\Theta^{(k)} = (\Sigma^{(k)})^{-1}$, where $\Sigma^{(k)}$ is the covariance matrix of observed variables under environment $k$. For a pair of environments $(1, 2)$, the difference $\Delta = \Theta^{(1)} - \Theta^{(2)}$ reveals the set of potentially affected nodes $S_\Delta = \{i : [\Delta]_{ii} \neq 0\}$. A node $i$ is identified as intervened if no subset $S \subseteq S_\Delta$ yields an invariant diagonal element $[\Delta_S]_{ii} = 0$, according to Lemma 2 of Varici et al. (2022). Similarly, parent and spouse sets of each intervened node are recovered via nonzero off-diagonal entries $[\Delta_S]_{ij} \neq 0$.

The method employs the precision-difference estimation (PDE) based on an ADMM solver with $\ell_1$-penalty:

$$\Delta^* = \arg\min_\Delta \frac{1}{2} \text{tr}(\Delta^\top \Sigma^{(1)} \Delta \Sigma^{(2)}) - \text{tr}(\Delta(\Sigma^{(1)} - \Sigma^{(2)})) + \lambda \|\Delta\|_1.$$

The resulting sparse estimate $\Delta^*$ is symmetrized and thresholded to determine the effective intervention targets $K$. The overall set of detected shifted nodes is defined as

$$S^{\text{PreDITEr}} := \{ j : \|\Delta_{j,\cdot}^*\|_1 > 0 \}.$$

### C.1.6. GPR

Gaussian Process Regression (GPR)-based Partial Permutation Test (Li et al., 2023) determines whether a shift in the conditional relationship between a node $X_j$ and its parent variables $\text{PA}_j$ is attributable to a functional shift or a noise shift. Under our implemented framework, we assume that a set of true shifted nodes $S^{\text{True}}$ has already been given. GPR is used as a follow-up test for classifying the nature of the shift. For each node $j \in S^{\text{True}}$, we model the structural equation in each environment $k \in \{1, 2\}$ as:

$$X_j^{(k)} = f_j^{(k)}(\text{PA}_j^{(k)}) + \varepsilon_j^{(k)},$$

where $f_j^{(k)}$ is a smooth (possibly nonlinear) function, and $\varepsilon_j^{(k)}$ is zero-mean Gaussian noise. We define four hypotheses modeling the relationship $X_j^{(k)} = f_j^{(k)}(\text{PA}_j^{(k)}) + \varepsilon_j^{(k)}$ across environments $k \in \{1, 2\}$:

1. **Null Model ($H_0$)** Assumes a shared function and shared noise variance across environments ($f^{(1)} = f^{(2)}$ and $\sigma_{(1)}^2 = \sigma_{(2)}^2$).

2. **Noise Shift Model ($H_{\textbf{noise}}$)** Assumes a shared function but allows environment-specific noise variances ($f^{(1)} = f^{(2)}$ and $\sigma_{(1)}^2 \neq \sigma_{(2)}^2$).

3. **Function Shift Model ($H_{\textbf{func}}$)** Assumes environment-specific functions but a shared noise variance ($f^{(1)} \neq f^{(2)}$ and $\sigma_{(1)}^2 = \sigma_{(2)}^2$).

4. **Simultaneous Shift Model ($H_{\textbf{simul}}$)** Assumes both environment-specific functions and environment-specific noise variances ($f^{(1)} \neq f^{(2)}$ and $\sigma_{(1)}^2 \neq \sigma_{(2)}^2$).

We compute the maximum log-likelihood estimates (MLE) for the data under each hypothesis, denoted as $\ell_0$, $\ell_{\text{noise}}$, $\ell_{\text{func}}$, and $\ell_{\text{simul}}$, respectively.

**Dissection (Function vs. Noise)** To distinguish between function shift and noise shift, we compare the likelihood improvements of $H_{\text{func}}$ and $H_{\text{noise}}$ over the null model $H_0$. We define the likelihood ratios as $\Delta_{\text{func}} = \ell_{\text{func}} - \ell_0$ and $\Delta_{\text{noise}} = \ell_{\text{noise}} - \ell_0$. More precisely, the classification rule is:

$$\text{Shift Type}_j = \begin{cases} \text{Function} & \text{if } \Delta_{\text{func}} > \Delta_{\text{noise}}, \\ \text{Noise} & \text{otherwise.} \end{cases}$$

That is, if separating functions yields a greater likelihood gain than allowing for variance heterogeneity, the shift is primarily functional.

## C.2. Simultaneous Shift

### C.2.1. FANS

To distinguish between function-only shifts and simultaneous shifts (where both function and noise distributions change), we employ a statistical testing procedure grounded in Theorem 3.4 of the main paper. The theorem posits that under a function-only shift (and satisfying Assumption 3.3), the squared standardized inferred noise must converge to a Chi-squared distribution with one degree of freedom, $\chi^2(1)$. Deviations from this theoretical distribution imply the presence of a simultaneous shift.

We implement this test using a non-parametric approach to estimate the conditional moments required for standardization. This procedure avoids strong parametric assumptions about the functional form of the shift and consists of the following three steps:

**1. Conditional Moment Estimation via GAMs** We utilize Generalized Additive Models (GAMs) to estimate the conditional mean and variance of the inferred noise $N_j^{(2)} = \hat{q}_j(\tilde{\epsilon}_j^{(2)})$ given the parent variables $PA_j^{(2)}$. First, we fit a GAM with spline terms to estimate the conditional expectation $\hat{\mu}_j(pa) \approx \mathbb{E}[N_j^{(2)} \mid PA_j^{(2)} = pa]$. To capture complex dependencies and interactions between multiple parent variables, we include tensor product terms for pairwise interactions in the GAM formulation. Subsequently, we compute the squared residuals $r^2 = (N_j^{(2)} - \hat{\mu}_j(PA_j^{(2)}))^2$ and fit a second GAM to estimate the conditional variance $\hat{\sigma}_j^2(pa) \approx \text{Var}(N_j^{(2)} \mid PA_j^{(2)} = pa)$. We enforce a minimum variance floor to ensure numerical stability during the subsequent standardization step.

**2. Noise Standardization** Using the estimated conditional moments, we compute the standardized noise $Z_j$ for each data point in Environment 2 as follows:

$$Z_j = \frac{N_j^{(2)} - \hat{\mu}_j(PA_j^{(2)})}{\sqrt{\hat{\sigma}_j^2(PA_j^{(2)})}} \tag{55}$$

Under the null hypothesis of a function-only shift, the square of this standardized variable, $Z_j^2$, follows a $\chi^2(1)$ distribution.

**3. Hypothesis Testing** Finally, we perform a Kolmogorov-Smirnov (KS) test to compare the empirical distribution of the squared standardized residuals $Z_j^2$ against the theoretical cumulative distribution function of $\chi^2(1)$.

- **Function-Only Shift:** If the p-value of the KS test is greater than the significance level $\alpha = 0.05$, we fail to reject the null hypothesis. We conclude that the standardized noise is consistent with $\chi^2(1)$, indicating the shift is fully explained by the functional change.

- **Simultaneous Shift:** If the p-value is less than $\alpha = 0.05$, we reject the null hypothesis. This indicates that the standardized noise deviates from the expected theoretical distribution, suggesting that the noise distribution itself has also shifted (Simultaneous Shift).

### C.2.2. GPR

To identify simultaneous shifts, we compare the function-only shift model ($H_{\text{func}}$) against the simultaneous shift model ($H_{\text{simul}}$). We evaluate whether adding noise heterogeneity to a function shift model significantly improves the fit. The decision rule is based on the comparison of $\Delta_{\text{func}}$ and $\Delta_{\text{simul}} = \ell_{\text{simul}} - \ell_0$:

$$\text{Shift Type}_j = \begin{cases} \text{Simultaneous} & \text{if } \Delta_{\text{simul}} > \Delta_{\text{func}}, \\ \text{Function} & \text{otherwise.} \end{cases}$$

That is, if the simultaneous model provides a higher likelihood gain, we can conclude that both the function and the noise distribution have been shifted.

### C.3. Morpho-MNIST

The Morpho-MNIST dataset extends the MNIST digits with quantitative morphometric attributes—thickness, length, width, height, and slant—to enable systematic analysis of representation learning (Castro et al., 2019). Building on its causal extensions, such as DeepSCM (Pawlowski et al., 2020) and ImageCFGen (Dash et al., 2022), we adopt a similar causal framework but modify it to focus exclusively on the digit '4'. By excluding the digit label ($l$) from the generative process, we isolate the effects of continuous morphological variables—thickness ($t$), intensity ($i$), and slant (s)—to examine their causal interactions within a controlled single-digit setting.

We define a causal graph over the image by treating each pixel as a node and assuming that all morphological attributes causally affect every pixel. We assume that the pixel-level nodes do not have causal relationships with one another. We crop the image borders to obtain a $20 \times 20$ resolution. We separate environment 1 and environment 2 based on the thickness attribute. We first min–max normalize thickness values to the range $[-1, 1]$, then assign samples with normalized thickness $\leq -0.5$ to environment 1 and those with thickness $\geq 0$ to environment 2. This preprocessing results in 2,645 images in environment 1 and 370 images in environment 2.

We use the normalized thickness for environment separation; all morphological variables are normalized using the mean and standard deviation of samples within each environment. By doing so, a same thickness value has a different meaning in different environments, leading to node shifts. For the image data, we preprocess the pixel nodes so that their values, originally ranging from 0 to 255 in the grayscale MNIST images, are centered around 127.5 and rescaled to the range $[-1, 1]$. We construct the adjacency matrix by ordering nodes according to the causal graph, placing root nodes first. Thus, we define the node order as $s, t, i, p_0, \ldots, p_{399}$.

FANS trains a normalizing flow on env1 ($n = 2645$, internal 80/20 train/validation split), then computes per-node Jensen–Shannon divergences on a evaluation pair of 370 samples drawn from each environment. For iSCAN, a single statistical two-sample test (no training phase) is applied to a pair of 370 samples drawn from each environment.

### C.4. Sachs protein signaling dataset

The Sachs dataset consists of flow cytometry measurements of 11 proteins and phospholipids in a human T-cell signaling pathway under multiple experimental perturbations.

**Shift detection criterion (Sachs).** For node-wise shift detection, we use the same detection-stage Jensen–Shannon divergence (JSD) statistic as in our general procedure, denoted by $\bar{\mathcal{D}}_j$ (the aggregated JSD computed in the detection stage). For the Sachs analysis, we utilize elbow-based selection used in synthetic experiments.

## C.5. Computations

All experiments were conducted on a workstation equipped with an Intel(R) Core(TM) i9-10900X CPU (20 cores) and $4\times$ NVIDIA GeForce RTX 4090 GPUs. We measured the wall-clock time for each method, representing the total duration required to process one configuration (consisting of 30 datasets). While parallel processing was utilized where feasible, GPR was executed serially to maintain efficiency, as processing a single node with GPR consumes over 70% of the entire CPU capacity. SplitKCI was also executed serially for efficiency.

*Table 4.* Runtime comparison for Shift Detection task on ER and SF graphs across varying node counts.

| Method | 10 Nodes | 20 Nodes | 30 Nodes | 40 Nodes | 50 Nodes |
|---|---|---|---|---|---|
| *ER graph* | | | | | |
| iSCAN | 17s | 29s | 42s | 55s | 1m 9s |
| SplitKCI | 4m 19s | 8m 36s | 12m 20s | 16m 35s | 20m 21s |
| LCIT | 3m 4s | 7m 11s | 10m 14s | 13m 44s | 17m 26s |
| LinearCCP | <1s | 1s | 2s | 3s | 3s |
| PreDITEr | 1s | 1s | 16s | 1m 42s | 37m 4s |
| FANS (Inference) | 6s | 13s | 18s | 33s | 43s |
| FANS (Train) | 189m 30s | 190m 32s | 193m 48s | 195m 54s | 197m 24s |
| *SF graph* | | | | | |
| iSCAN | 16s | 29s | 41s | 55s | 1m 9s |
| SplitKCI | 6m 28s | 13m 37s | 20m 59s | 28m 19s | 35m 38s |
| LCIT | 7m 7s | 15m 7s | 23m 23s | 31m 20s | 40m 7s |
| LinearCCP | <1s | <1s | <1s | <1s | <1s |
| PreDITEr | <1s | 2s | 13s | 3m 4s | 7m 54s |
| FANS (Inference) | 5s | 13s | 28s | 43s | 52s |
| FANS (Train) | 189m 38s | 191m 24s | 194m 35s | 196m 1s | 197m 38s |

*Table 5.* Runtime comparison for Shift Dissection task on ER and SF graphs across varying node counts.

| Method | 10 Nodes | 20 Nodes | 30 Nodes | 40 Nodes | 50 Nodes |
|---|---|---|---|---|---|
| *ER graph* | | | | | |
| iSCAN | <1s | <1s | <1s | <1s | 1s |
| LinearCCP | <1s | 1s | 2s | 3s | 3s |
| GPR | 97m 39s | 285m 14s | 314m 48s | 557m 30s | 486m 51s |
| FANS (Inference) | <1s | <1s | <1s | 1s | 2s |
| *SF graph* | | | | | |
| iSCAN | <1s | <1s | <1s | <1s | <1s |
| LinearCCP | <1s | <1s | <1s | <1s | <1s |
| GPR | 90m 19s | 252m 2s | 298m 54s | 359m 56s | 407m 6s |
| FANS (Inference) | <1s | <1s | <1s | <1s | <1s |

# D. Additional Analyses and Ablations

## D.1. Training Hyperparameter and Threshold Sensitivity

### D.1.1. HYPERPARAMETER SENSITIVITY

We additionally report the per-sample log-likelihood of the trained CNF, evaluated at each run's best-validation checkpoint and averaged over 30 random datasets per variant in Table 6. We used this quantity to select hyperparameters for the synthetic experiments. Note that the log-likelihood only measures how well the CNF fits the observational data and is therefore not a direct evaluation of how well each option performs at detection or dissection.

Table 7 demonstrates that FANS is robust to hyperparameter choices of batch size and learning rate: F1 scores vary

marginally across 8 configurations.

*Table 6.* Validation loss across flow architectures and training hyperparameters. Each cell reports mean validation loss over 30 DAGs. Lower is better. The best configuration is highlighted in **bold**.

| Flow | LR | Batch Size | | | |
|------|------|-------|-------|-------|-------|
| | | 512 | 1024 | 2048 | 4096 |
| NSF | 0.01 | **56.02** | 56.09 | 56.15 | 56.28 |
| | 0.001 | 56.58 | 56.80 | 56.99 | 57.17 |
| NAF | 0.01 | 57.27 | 57.70 | 57.23 | 57.35 |
| | 0.001 | 57.34 | 57.15 | 58.08 | 58.60 |

*Table 7.* Hyperparameter sensitivity for FANS across flow architectures (NSF, NAF) on 50-node ER graphs. Each cell reports F1 (Detection) / F1 (Dissection). Hidden layers fixed at [32,32,32], 3000 epochs. The default configuration is highlighted in bold.

| Flow | LR | Batch Size | | | |
|------|------|-------|-------|-------|-------|
| | | 512 | 1024 | 2048 | 4096 |
| NSF | 0.01 | **0.975 / 0.947** | 0.975 / 0.945 | 0.971 / 0.951 | 0.976 / 0.940 |
| | 0.001 | 0.964 / 0.935 | 0.964 / 0.928 | 0.966 / 0.936 | 0.962 / 0.921 |
| NAF | 0.01 | 0.942 / 0.883 | 0.928 / 0.891 | 0.932 / 0.882 | 0.920 / 0.899 |
| | 0.001 | 0.925 / 0.879 | 0.920 / 0.882 | 0.916 / 0.872 | 0.908 / 0.872 |

### D.1.2. DETECTION AND DISSECTION THRESHOLDS

*Table 8.* Threshold sensitivity on 50-node ER graphs.

| Task | Threshold value | F1 Score |
|------|------|------|
| Detection ($\tau$) | 0.06 | 0.965 |
| | 0.08 | 0.995 |
| | 0.10 | 0.975 |
| | 0.12 | 0.921 |
| | 0.14 | 0.835 |
| Dissection | 0.02 | 0.937 |
| | 0.04 | 0.941 |
| | 0.06 | 0.935 |
| | 0.08 | 0.906 |

Table 8 reports F1 scores on 50-node ER graphs across a range of plausible values. FANS relies on two thresholds: (i) a divergence threshold $\tau$ used in the detection stage to classify a node as shifted if its aggregated divergence $\bar{\mathcal{D}}_j$ exceeds $\tau$, and (ii) a distance correlation threshold used in the dissection stage to classify a shifted node as a function shift versus a noise shift. While our main experiments use an elbow-based procedure for $\tau$ and a fixed default of $0.05$ for the dissection threshold, we here examine the sensitivity of FANS to manual choices of both thresholds.

## D.2. Robustness to Sample Size and Model Capacity

### D.2.1. SAMPLE SIZE

We evaluate how FANS performance scales with training sample size. Using the 50-node ER graph setting with the default configuration, we reduce the number of samples per environment from $50,000$ to $5,000$, holding out $20\%$ of the data for validation in each case.

*Table 9.* FANS performance under varying sample sizes on 50-node ER graphs. Performance degrades gracefully as the training set shrinks.

| Samples $N$ | F1 (Detection) | F1 (Dissection) |
|---|---|---|
| 50,000 | 0.975 | 0.947 |
| 30,000 | 0.971 | 0.941 |
| 10,000 | 0.973 | 0.938 |
| 5,000 | 0.970 | 0.916 |

FANS maintains strong performance even when the training set is reduced by $80\%$. Detection F1 drops only modestly from $0.975$ to $0.970$, and dissection F1 remains above $0.91$ throughout. This indicates that our framework is viable in practical settings where large observational datasets may not be available.

### D.2.2. TRAINING EPOCHS

*Table 10.* FANS performance under varying training epochs on 50-node ER graphs. Performance remains strong even at 100 epochs.

| Epochs | F1 (Detection) | F1 (Dissection) |
|---|---|---|
| 100 | 0.966 | 0.937 |
| 500 | 0.977 | 0.940 |
| 3000 | 0.975 | 0.947 |

We next examine sensitivity to training duration by reducing the number of epochs from our default of 3000. Even with 100 epochs, detection F1 remains above $0.96$ and dissection F1 above $0.93$. This suggests that FANS does not require extensive training to achieve reasonable performance, which is useful when computational budgets are constrained.

### D.2.3. MODEL CAPACITY

To assess robustness to model misspecification via underparameterization, we reduce the CNF's hidden layer dimensions below the default of $[32, 32, 32]$.

*Table 11.* FANS performance under varying CNF hidden layer architectures on 50-node ER graphs. Unlike training hyperparameters, model capacity has a pronounced effect on performance.

| Hidden Layers | F1 (Detection) | F1 (Dissection) |
|---|---|---|
| $[32, 32, 32]$ | 0.975 | 0.947 |
| $[16, 16]$ | 0.917 | 0.864 |
| $[8, 8]$ | 0.871 | 0.794 |

In contrast to the relative insensitivity to batch size, learning rate, and training duration, model capacity plays a more significant role. Reducing the hidden dimensions to $[8, 8]$ leads to a noticeable drop in both detection and dissection performance, indicating that sufficient network expressiveness is required to capture the non-linear, non-additive structural equations considered in our framework. This aligns with the expected behavior of flow-based models: the CNF must be expressive enough to invert the underlying data-generating process, and insufficient capacity propagates estimation error into both the divergence statistic and the residual independence test.

### D.3. Robustness to Assumption Violations

#### D.3.1. VIOLATION OF ASSUMPTION

Assumption 3.3 (Generalized Affine Interaction) underpins the identifiability result between function-only and simultaneous shifts (Theorem 3.4). To evaluate robustness when this assumption is violated, we introduce a quadratic noise term controlled

by a severity parameter $\lambda$, modifying the data-generating process as:

$$X_j = \sum_k \sin([\mathrm{PA}_j]_k^2) + \sum_k \sigma([\mathrm{PA}_j]_k^2) \cdot \varepsilon_j + \lambda \left( \sum_k [\mathrm{PA}_j]_k^2 \right) \cdot \varepsilon_j^2, \tag{56}$$

where $\lambda = 0$ recovers our original setting and larger values induce stronger violations of the generalized affine interaction structure. As expected, performance degrades under assumption violations, since the standardized-noise argument in Theorem 3.4 no longer holds exactly.

*Table 12.* FANS performance for distinguishing function-only from simultaneous shifts under violations of Assumption 3.3. The severity parameter $\lambda$ controls the magnitude of the violation.

| Severity $\lambda$ | F1 Score |
|---|---|
| 0.0 (no violation) | 0.622 |
| 0.5 (mild) | 0.333 |
| 1.0 (severe) | 0.310 |

### D.3.2. ADDITIVE NOISE MODELS

Although FANS is designed for a broader class of non-additive structural equations, it should remain valid under the more restricted Additive Noise Model (ANM) setting. We evaluate this by setting the scale term $\sigma(\cdot) = 1$ in the data-generating process of Section 4.1 on 50-nodes ER graphs, reducing the SCM to an ANM.

*Table 13.* FANS performance on 10-node ER graphs under the ANM setting ($\sigma(\cdot) = 1$). Standard deviations across 30 DAGs are reported in parentheses.

| | F1 (Detection) | F1 (Dissection) |
|---|---|---|
| FANS (ANM setting) | 0.909 (0.084) | 0.822 (0.206) |

FANS achieves a moderate detection performance under the ANM setting. We attribute this to the fact that removing the parent-dependent scale term weakens the distributional signal distinguishing noise shifts from function shifts: when noise enters purely additively, a change in noise distribution produces a more subtle effect on the conditional distribution, making dissection inherently harder. This suggests that FANS benefits from heteroskedastic noise structure to perform dissection reliably.

### D.3.3. ENVIRONMENT SWAP

FANS assumes that the causal DAG of the base environment (Environment 1) is known. To examine whether the choice of base environment affects performance, we swap the roles of Environments 1 and 2 on 15 datasets with 50-node ER graphs where the DAG structure is preserved across environments. In these cases, FANS achieves an F1 score of $0.953$ for detection and $0.979$ for dissection. This indicates that, when the structural graph is shared, the assignment of base environment is not a source of bias in our procedure.

### D.3.4. UNOBSERVED CONFOUNDING

Our framework assumes causal sufficiency, the absence of unobserved common causes. We briefly discuss the behavior of FANS when this assumption is violated. Consider an unobserved confounder $U$ affecting both $X$ and $Y$ ($U \to X$, $U \to Y$, $X \to Y$). If the distribution of $U$ shifts across environments, FANS will capture this variation through the observed residuals: the shift in $U$ typically manifests as an apparent simultaneous mechanism shift on $Y$ (as $U$ enters $Y$'s structural equation) together with a noise shift on $X$ (as the marginal of $X$ changes). While this behavior does not recover the true underlying cause, the pattern could serve as a diagnostic signal for the presence of shifting hidden confounders.

