# OpenReview forum: "Dissecting Causal Mechanism Shifts via FANS: Function And Noise Separation"
_ICML.cc/2026/Conference — ICML 2026 regular_

### Official Review · Reviewer_U6rR · 2026-02-26

**Soundness:** 3
**Presentation:** 3
**Significance:** 2
**Originality:** 3
**Overall Recommendation:** 4
**Confidence:** 4

**Summary:**

Assuming a class of structural causal models (SCMs) and observations from two environments, the paper studies how to detect changes in causal mechanisms and distinguish whether they occur in the structural functions (i.e., the relationships between variables and their parents) or in the exogenous noise distributions. The considered SCM class generalizes additive noise models by allowing invertible structural equations of the form $g_j(h_j(PA_j), \varepsilon_j))$, where the link function $g_j$ is invariant across environments. To detect causal changes, the authors fit a causal normalizing flow (CNF) to data from the first environment and compare conditional distributions across environments using statistical tests. To determine the type of change, they test independence between predicted residuals and parent variables in the second environment.

**Compliance With Llm Reviewing Policy:**

Affirmed.

**Final Justification:**

My main concerns have been addressed.

**Key Questions For Authors:**

1. What threshold $\tau$ for detection and threshold for the independence test were used in the experiments in Section 4, and why? I expect the results to be different if different value thresholds are used. Please also provide these results.
2. Do the authors perform any sample splitting for fitting and evaluating the conditional distributions in environment 1?
3. How does FANS compare to other existing work when the true model is an additive noise model, since FANS is more general but should still work under more restricted models?
4. Can the authors discuss or show how FANS perform where there exists unobserved confounding, and the confounder has a distribution shift between the two environments?
5. What is  FCI-JCI123 in Section 4.4?

**Limitations:**

Limitations were not addressed in the paper at all so far. I list some limitations below.
1. This work assumes the knowledge of the causal graph, but e.g., Huang et al. (2024) cited in Section 1.1 does not require this knowledge.  This difference was not made clear in the paper. If I’m not mistaken, only until Section 2 did the authors mention that `` Given the data from both environments and the causal DAG from the base environment, our goal is […]”.
2. The authors should also address the consequences of the assumptions. For example, Assumption 2.4 means that interventions cannot result in a node having more parents than before the intervention. This seems to be the reason why, under equation (4), ``We assume that the causal structure in environment 2 results from interventions on $\mathcal{G}(1)$. It implies that the parent set for each variable Xj in environment 2 is a subset of the parent set for Xj in environment 1”.
3. The authors should connect Assumption 2.6 to the independent causal mechanisms and modularity assumption.
4. The authors should explain how the threshold values are chosen in practice.

**Strengths And Weaknesses:**

Strength:

- [Presentation] The paper is relatively well written.
- [Originality] The extension from additive noise models to a possibly non-additive but invariant function in causal change point detection is new in the literature, to my knowledge.
- [Soundness] The theoretical results look sound to me.

Weakness:

- [Presentation] I find the experiment section unclear. For example, how were the Morpho-MNIST variables and the DAG decided (e.g., why are thickness and intensity two variables)? The message of Section 4.4 is also unclear to me, so FANS identifies 5 out of 11 variables that have a causal mechanism change, although only one (Mek) was intervened on, which has no descendants.
- [Significance] The paper assumes the knowledge of a DAG in the first environment, which I don’t think is realistic in applications. In the absence of such a DAG, one needs to first perform causal discovery in the first environment, which may already have errors.

---

> ### Author Rebuttal · Authors · 2026-03-31
>
> > W1 Presentation
>
> Morpho-MNIST: The DAG structure and the selection of variables (slant, thickness, intensity) were adopted directly from Taylor-Melanson et al. (2024). We will explicitly cite this source and clarify the graph construction in the revised text.
>
> Sachs Dataset (Section 4.4): In Environment 2, the Mek protein is inhibited by the reagent U0126. Because different causal discovery methods yield different consensus graphs regarding the exact downstream effects of this inhibition (e.g., Mooij et al. identify Erk as a child; FCI-JCI identifies Mek, Erk, and PKC; Eaton & Murphy identify a broader set including RAF, Mek, Erk, Akt, PKC, and JNK), our goal was to demonstrate which nodes experience a shift in their generative mechanisms when U0126 is introduced.
>
> > W2 Significance
>
> We made this assumption because the primary focus of our work is the dissection of causal mechanism shifts, which fundamentally requires structural knowledge to isolate parent-child relationships. We acknowledge that relying on a pre-defined or perfectly discovered DAG is a limitation in practice. We will state this assumption as a limitation in the revised manuscript.
>
> > Q1 Threshold Selection
>
> Regarding the selection strategy, please refer to the answer of Q1 from sEr5. To address your concern regarding sensitivity, we conducted an ablation study by manually varying both thresholds. We intentionally avoid over-optimizing these thresholds on the training set to ensure robust generalization. We will include this complete sensitivity analysis in the revised appendix.
>
> **Detection Threshold ($\tau$) Sensitivity:**
>
> | Threshold | 0.06 | 0.08 | 0.10 | 0.12 | 0.14 |
> | :--- | :--- | :--- | :--- | :--- | :--- |
> | **F1 Score** | 0.9356 | 0.9800 | 0.9933 | 0.9889 | 0.9444 |
>
> **Dissection Threshold Sensitivity:**
>
> | Threshold | 0.10 | 0.12 | 0.14 | 0.16 |
> | :--- | :--- | :--- | :--- | :--- |
> | **F1 Score** | 0.9778 | 1.0000 | 1.0000 | 0.9778 |
>
> > Q2 Sample Splitting
>
> Out of the 50,000 data points generated per environment, 40,000 were used for actual training of CNF, and 10,000 were held out as a validation set. This split was strictly used to monitor the learning curve.
>
> > Q3 FANS under ANM
>
> For an ANM setting, we set the scale term $\sigma(\cdot) = 1$ in the equation of Section 4.1. FANS achieves an F1 score of 0.922 ± 0.143 for detection and 0.844 ± 0.273 for dissection on 10-node ER graphs. The dissection score is slightly lower, which may be attributed to the fact that removing the heteroskedastic scale term weakens the signal distinguishing noise shifts from function shifts, making dissection inherently harder in the additive setting.
>
> > Q4 Unobserved Confounder
>
> Let an unobserved confounder $U$ affect both $X$ and $Y$ ($U \to X$, $U \to Y$, and $X \to Y$). If the distribution of $U$ shifts across environments, FANS will capture this variation and flag it as a simultaneous mechanism shift in $Y$ and a noise shift in $X$. FANS can potentially serve as a diagnostic indicator for shifting hidden confounders, which would be an interesting direction for future work.
>
> > Q5 What is FCI-JCI123 in Section 4.4?
>
> FCI-JCI123, introduced by Mooij et al., 2020, is a variant of the FCI-JCI algorithm to optimize the inference process. It is a causal discovery framework designed to learn causal graphs from multiple contexts or environments simultaneously.
>
> > L1 DAG Assumption.
>
> We appreciate this observation. It is true that some detection methods (e.g., iSCAN) can identify shifted nodes without requiring a fully known DAG. However, for the dissection task, all baselines rely on knowledge of the parent sets. This is because dissection fundamentally requires isolating the relationship between a node and its specific parents to determine the source of the shift. In this regard, the DAG requirement is not unique to FANS but is an inherent condition of the dissection problem itself. That said, we will add a clear statement in the Conclusion section.
>
> > L2 Consequence of Assumption 2.4.
>
> Assumption 2.4 requires that the link function $g_j$ is invariant across environments, so all cross-environment variation is absorbed by changes in $h_j$. Because $g_j$ is shared, the parent set in Environment 2 can only remain the same or shrink (via edge deletion), but cannot grow. This is precisely why Equation (4) requires the parent set to be a subset. We will make this connection explicit in the revised manuscript.
>
> > L3 Assumption 2.6.
>
> Assumption 2.6 is indeed a direct formalization of the Principle of Independent Causal Mechanisms and Modularity. We will update the text to explicitly state this connection, situating our assumption firmly within standard causal inference literature.
>
> > L4 Threshold selection.
>
> A detailed description and sensitivity analysis are provided in our response to Q1 of Reviewer sEr5.

---

> > ### Author Rebuttal · Reviewer_U6rR · 2026-04-02
> >
> > I appreciate the authors' response and providing the additional results. I keep my original score.

---

> > > ### Author Response · Authors · 2026-04-03
> > >
> > > We sincerely thank you for confirming that your concerns have been fully resolved. We will make sure the additional results and revisions are carefully incorporated into the camera-ready version. Thank you again for your constructive feedback, which has greatly helped strengthen our paper.

---

### Official Review · Reviewer_oGUv · 2026-03-10

**Soundness:** 3
**Presentation:** 3
**Significance:** 3
**Originality:** 3
**Overall Recommendation:** 3
**Confidence:** 2

**Summary:**

This paper introduces the FANS framework for detecting and dissecting causal mechanism shifts (CMS) across environments in structural causal models. The key objective is to distinguish whether a shift in a conditional distribution arises from a change in the causal function or from a change in the exogenous noise distribution. The approach leverages Causal Normalizing Flows (CNFs) trained on a base environment to approximate invertible mappings from observed variables to noise variables. Shift detection is conducted via divergence between counterfactual distributions generated through do-operations under the learned CNF. Dissection is performed using an independence test between inferred residual noise and parent variables. The framework also discusses the non-identifiability between function-only and simultaneous function–noise shifts and proposes structural assumptions under which identifiability can be restored.

**Compliance With Llm Reviewing Policy:**

Affirmed.

**Key Questions For Authors:**

1. How sensitive is FANS to misspecification or approximation error in the learned CNF mapping? Have the authors evaluated performance when the CNF is deliberately underparameterized?
2. Can FANS be extended to settings where structural equations are not invertible? e.g., discrete nodes or piecewise-constant functions?
3. How is the divergence threshold $\tau$ selected in practice? Is there a theoretically grounded approach to controlling false discovery rates across nodes?
4. Are there bounds or asymptotic guarantees for the independence-based classification under estimation error?
5. The moderate F1 in Table 3 suggests difficulty in separating function-only from simultaneous shifts. Under what conditions can we expect reliable performance in realistic datasets?

**Limitations:**

The framework assumes known causal structure in the base environment and relies on invertible structural equations. It does not explicitly address latent confounding or causal structure misspecification. Additionally, independence-based tests may suffer from reduced power in high-dimensional parent sets. A more explicit discussion of these limitations would improve transparency.

**Strengths And Weaknesses:**

Strengths:
1. The distinction between function shifts and noise shifts is conceptually well articulated and motivated with intuitive examples.
2. By employing CNFs, the framework extends dissection to nonlinear and non-additive SCMs, relaxing assumptions common in prior work such as ANM-based methods.
3. The independence criterion (Theorem 3.2) provides a principled statistical characterization of shift types. The analysis of non-identifiability and the introduction of Assumption 3.3 reflect careful theoretical consideration.
4. Training only once on the base environment and applying inference to new environments without retraining is computationally appealing.
5. In synthetic experiments, FANS consistently achieves superior F1-scores in both shift detection and dissection compared to existing baselines.

Major Weaknesses:
1. Assumption 2.2 requires each structural equation to be a diffeomorphism in the noise variable. In many real-world SCMs (e.g., discrete outcomes or threshold mechanisms), this assumption may not hold. The practical robustness of FANS to approximate invertibility is not thoroughly analyzed.
2. The method relies critically on accurate estimation of the invertible mapping $\hat{q}$. Finite-sample CNF estimation errors may introduce spurious dependence or mask true shifts. The paper does not quantify how approximation error propagates to the independence test.
3. The detection rule $ \bar{D}_j > \tau $ depends on a predefined threshold $\tau$, but guidance on threshold calibration is limited. Similarly, the independence testing step relies on distance correlation; Type I error control under model misspecification is not extensively discussed.
4. The identifiability result (Theorem 3.4) hinges on Assumption 3.3 (Generalized Affine Interaction). While mathematically clear, its practical plausibility across domains is not fully justified. It would be helpful to clarify in which classes of real SCMs this assumption is expected to hold.
5. Although the paper compares against iSCAN, LinearCCP, and PreDITEr, it does not benchmark against more recent neural conditional distribution tests or representation-based shift detection methods.

Minor Weaknesses:
1. The use of 50,000 samples in synthetic experiments may obscure finite-sample behavior. It would be informative to see performance under smaller sample sizes.
2. In the simultaneous-shift experiment (Table 3), performance is moderate (F1 ≈ 0.598), suggesting that practical disentanglement remains challenging.
3. The Morpho-MNIST experiment provides qualitative visualization, but quantitative metrics for localization accuracy would strengthen the evaluation.

---

> ### Author Rebuttal · Authors · 2026-03-31
>
> We sincerely thank the reviewer for the constructive feedback and detailed evaluation.
>
> > Major W1, W4, Q2: Plausibility of Assumptions and Non-invertible extensions
>
> [W1] We agree with the reviewer that the diffeomorphism requirement is a limitation that restricts our framework's applicability. However, within the continuous domain, FANS captures a broader class of non-linear structural equations than standard Additive Noise Models (ANM). Furthermore, current state-of-the-art CNF-based methods [1-5] universally rely on this assumption.
>
> [W1, W4] In practice, real-world SCMs such as location-scale noise models satisfy Assumption 3.3 and Assumption 2.2. For example, the Engel curve in economics [6]:
>
> $$\text{Expenditure} = f(\text{Income})+\sigma (\text{Income})\cdot \varepsilon$$
>
> [Q2] Regarding discrete or piecewise nodes, extending FANS to discrete settings would require dequantization techniques, which is an exciting avenue for future work.
>
> References
> [1] Javaloy et al. NeurIPS, 2023.
> [2] Zhou et al. AAAI, 2025.
> [3] Balgi et al. AAAI, 2022.
> [4] Cho & Sun. Mathematics, 2025.
> [5] Xu et al. IJCAI, 2025.
> [6] Koenker & Hallock. J. Econ. Perspect., 2001.
>
> >Major W2, W3, Q3, Q4: Estimation Errors
>
> [Q3] Divergence threshold $\tau$ can be selected via an elbow method(please see the answer of Q1 of reviewer sEr5)
>
> [Major W2, W3, Q4]
> We have formalized the bounds of FANS under CNF estimation error. Let $Z^{(k)} := PA_j^{(k)}$, $N^{(k)}$ be the true residual noise, and $\hat{N}^{(k)} := \hat{q}_j(\tilde{\epsilon}_j^{(k)})$ be the estimated noise. Let $(N', \hat{N}', Z')$ and $(N'', \hat{N}'', Z'')$ be i.i.d. copies.
> * **Assumption 1 :** $\mathbb{E}[|\hat{N}^{(k)} - N^{(k)}|] \le \delta_n$, where $\lim_{n \to \infty} \delta_n = 0$.
> * **Assumption 2 :** $\mathbb{E}[\|Z^{(k)} - Z^{(k)'}\|] = C_Z < \infty$.
>
> The error in Distance Covariance $\mathcal{V}^2$ is bounded: $|\mathcal{V}^2(\hat{N}, Z) - \mathcal{V}^2(N, Z)| \le 8 C_Z \delta_n$. By definition, $\mathcal{V}^2(N, Z)$ consists of three expectation terms.
> 1.  $\left| \mathbb{E}[|\hat{N}-\hat{N}'|\|Z-Z'\|] - \mathbb{E}[|N-N'|\|Z-Z'\|] \right| \le 2 C_Z \delta_n$
> 2.  $\left| \mathbb{E}[|\hat{N}-\hat{N}'|]\mathbb{E}[\|Z-Z'\|] - \mathbb{E}[|N-N'|]\mathbb{E}[\|Z-Z'\|] \right| \le 2 C_Z \delta_n$
> 3.  $2\left| \mathbb{E}[|\hat{N}-\hat{N}'|\|Z-Z''\|] - \mathbb{E}[|N-N'|\|Z-Z''\|] \right| \le 4 C_Z \delta_n$
> Summing these bounds gives $8 C_Z \delta_n$.
>
> Let $\hat{\mathcal{R}}_k$ denote the estimated distance correlation between $\hat{N}^{(k)}$ and $Z^{(k)}$. The test statistic offsetting the baseline correlation is : $\hat{S} = |\hat{\mathcal{R}}_2 - \hat{\mathcal{R}}_1|$, compared to the true statistic $S^{\*} = |\mathcal{R}_2 - \mathcal{R}_1|$.  Then, $|\hat{S} - S^{\*}| \le \Delta_n$, where $\Delta_n = \mathcal{O}(\sqrt{\delta_n})$.
>
> *Proof.* There exists a constant $K_k$ such that $|\hat{\mathcal{R}}_k - \mathcal{R}_k| \le K_k \sqrt{\delta_n}$. Applying the triangle inequality to the differential statistic:
> $$|\hat{S} - S^{\*}| \le |\hat{\mathcal{R}}_1 - \mathcal{R}_1| + |\hat{\mathcal{R}}_2 - \mathcal{R}_2| \le (K_1 + K_2)\sqrt{\delta_n} := \Delta_n$$
>
> Given a rejection threshold $\tau_\alpha$,
> - Type 1 Error (Pure Noise Shift, $S^{\*}=0$), $P(\hat{S} > \tau_\alpha \mid H_0) \le P(\Delta_n > \tau_\alpha) \to 0 \text{ as } n \to \infty$
> - Type 2 Error (Function Shift, $S^{\*}=c>\tau_\alpha$). $P(\hat{S} \le \tau_\alpha \mid H_1) \le P(\Delta_n \ge c - \tau_\alpha) \to 0$ as $n \to \infty$
>
> >Major W5
>
> We have added the Latent Representation-based Conditional Independence Test (LCIT, ICDM 2022). For the detection task (ER graph, 10 nodes), LCIT achieved an F1 score of 0.548 ± 0.354.
>
> >Minor W1 & Q1
>
> Our default setting (Section 4.1) uses [32,32,32] hidden layers and 3000 epochs. We re-ran FANS on smaller subsets of the 10-node ER dataset (with 20% held out for validation in each case).
>
> Samples, F1 score (detection), F1 score (dissection)
> - N=50,000 | 0.993 | 0.978
> - N=30,000 | 0.993 | 0.956
> - N=10,000 | 0.969 | 0.956
>
> We also tested with reduced training epochs.
> Number of Epochs, F1 score (detection), F1 score (dissection)
> - 100 | 0.980 | 0.922
> - 500 | 0.982 | 0.956
> - 3000 | 0.993 | 0.978
>
> We further tested with underparameterized CNFs by reducing the hidden layer dimensions.
>
> Hidden layer architecture: F1 score (detection), F1 score (dissection)
> - [32,32,32]: 0.993 | 0.978
> - [16,16]: 0.953 | 0.944
> - [8,8]: 0.767 | 0.700
>
> > Minor W2 & Q5
>
> Our moderate F1 score (0.598) represents a better performance than the baseline for this challenging disentanglement task. Reliable performance is expected when the conditional moment estimation (via GAMs) is stable.
>
> >Minor W3
>
> We computed the exact pixel-wise difference between the empirical averages of Environment 1 and Environment 2, extracting the top 100 pixels with the largest true shifts. FANS correctly detected 77 of these ground-truth pixels, whereas iSCAN detected only 2.

---

> > ### Author Rebuttal · Reviewer_oGUv · 2026-04-03
> >
> > Thanks for the detailed rebuttal. The additional experiments on smaller samples and underparameterized CNFs are helpful, and the provided error analysis gives some theoretical insight into how estimation errors affect the independence test.

---

> > > ### Author Response · Authors · 2026-04-04
> > >
> > > We sincerely thank for the constructive feedback. We are very glad to hear that our responses have successfully addressed all of your concerns. In light of this, we respectfully ask if you might consider raising your score. Please do not hesitate to let us know if you have any further questions during the discussion period.

---

### Official Review · Reviewer_zwKH · 2026-03-13

**Soundness:** 3
**Presentation:** 3
**Significance:** 3
**Originality:** 3
**Overall Recommendation:** 5
**Confidence:** 3

**Summary:**

This submission proposes Function and Noise Separation (FANS), a framework to detect and dissect shifts in non-additive, non-linear Structural Causal Models (CSMs). The authors build a two-stage algorithm to detect shifts in statistical dependence between nodes and noise, and achieve some identifiability with a new assumption on simultaneous function and noise shifts.

**Compliance With Llm Reviewing Policy:**

Affirmed.

**Final Justification:**

This problem of causal attribution is meaningful. I am positive about this submission, and the rebuttal has reinforced my positive assessment.

**Key Questions For Authors:**

None

**Limitations:**

Yes

**Strengths And Weaknesses:**

- _Soundness_: The submission appears technically sound. I did not carefully read all proofs in the appendix, but the framework appears reasonable. Assumption 3.3 (Generalized Affine Interaction) seems helpful, and the resulting Theorem 3.4 (Shift Identifiability) gives confidence in recovering some causal effects. The empirical results are limited in scope but strong in effect across simulation and two real-world datasets. Recovery of the function shift in Figure 4 is quite motivating.
   - What are the values displayed in parentheses in Tables 1 and 2? While not described, I assume these are variances of the F1-scores; the low variance for the FANS method is notable.
- _Presentation_: The paper is presented well, with the method positioned fairly to the best of my knowledge.
- _Significance_: Causal discovery is an important scientific discovery. With heterogeneous datasets growing, the capacity to discover causal structure under shift is ever more useful.
- _Originality_: The method is original to the best of my knowledge.

---

> ### Author Rebuttal · Authors · 2026-03-31
>
> We sincerely thank the reviewer for the positive evaluation and for recognizing the soundness, originality, and significance of our framework.
>
> To clarify the reviewer's observation: the values displayed in parentheses in Tables 1 and 2 are indeed standard deviations of the F1-scores across 30 random DAGs. We appreciate the reviewer noting that the low variance for FANS is a notable strength, reflecting the stability of our method across diverse graph structures.
>
> In response to concerns raised by other reviewers, we have substantially strengthened the paper with additional experiments and analyses that we believe further support the contributions recognized by the reviewer. Specifically, we (1) added a comprehensive hyperparameter sensitivity analysis demonstrating robustness across batch sizes and learning rates, (2) evaluated FANS under smaller sample sizes (down to N=10,000) and under-trained models (100 epochs), showing graceful performance degradation, (3) formalized theoretical error bounds quantifying how finite-sample CNF approximation error propagates to the independence test statistic, (4) added a quantitative localization metric for Morpho-MNIST (FANS correctly detected 77 of the top 100 shifted pixels vs. only 2 for iSCAN), and (5) included an additional baseline (LCIT) for broader comparison. We believe these additions further corroborate the empirical strength and theoretical grounding that the reviewer identified in our original submission.

---

> > ### Author Rebuttal · Reviewer_zwKH · 2026-04-04
> >
> > Thank you for the response. I maintain my positive score.

---

> > > ### Author Response · Authors · 2026-04-07
> > >
> > > We sincerely thank you for taking the time to review our work and for your highly positive evaluation. Your encouraging feedback is deeply appreciated.

---

### Official Review · Reviewer_sEr5 · 2026-03-19

**Soundness:** 4
**Presentation:** 3
**Significance:** 3
**Originality:** 4
**Overall Recommendation:** 5
**Confidence:** 5

**Summary:**

In this work, the authors study whether one can automatically detect changes in the causal mechanisms between related environments, distinguishing between changes in the causal generator (function shift) and the exogenous distribution (noise shift). To this end, the authors propose to train a causal normalizing flow on one of the environments, and then evaluate it on the other one: If the two distributions diverge significantly, one can run a statistical test to see whether the parents are exogenous variable of each variable are independent. If they are, then the shift was noise shift, otherwise it was a functional and (potentially) noise shift. Then the authors discuss under which conditions once can detect these simultaneous shifts, and then evaluate and compare their model in synthetic as well as real data.

**Compliance With Llm Reviewing Policy:**

Affirmed.

**Final Justification:**

I think this is a rather solid paper addressing a not-so-explored problem for which the authors propose a nice solution that leverages causal generative models and which seems to significantly beat prior approaches.

My main concerns regarded the lack of experimental details and the evaluation on the same data-generating processes assumed in the theory. During the rebuttal both of these questions were addressed, and I trust the authors will add to the final version every single experimental detail needed to reproduce their results.

**Key Questions For Authors:**

- Q1. How do you choose the thresholds required in the paper?
- Q2. How do you know which environment is environment 1 and which one is 2? Couldn't you run the same experiment swapping them?
- Q3. I want to challenge the authors regarding the importance of the ambiguous case discussed in line 182. I am pretty confident that $\delta$ in line 174 cannot depend on the parents for that equation to hold. Then, does it really matter if I change the equation by a constant term or the noise itself?
- Q4. For the case where there is potentially a simultaneous change: Couldn't you train a second flow on env. 2 and then plug in the exogenous variables of environment 1 to see if there is on top of function shift a noise shift?
- Q5. Am I missing something or algorithm 1 and figure 2 do not reflect the cases where there are simultaneous shifts?
- Q6. What is the difference between tables 4 and 5? and what do those times measure exactly?

**Limitations:**

yes

**Strengths And Weaknesses:**

**Strengths:**
- S1. Personally, I find the idea of leveraging causal normalizing flows to dissect changes between environments refreshing and of interest to the community.
- S2. The paper is generally nicely written and easy to follow.
- S3. While there are more assumptions than those from the causal normalizing flow paper, they look rather reasonable and not too difficult to fulfill, except for maybe assumption 3.3.
- S4. While I would like the experiments on real datasets to exhibit quantitative performance, I think the qualitative results do a good job at reflecting the effectiveness of the proposed method.

**Weaknesses:**
- W1. While the paper is generally well redacted, it would still benefit from some polishing. For example, $\Psi$ is never introduced in the main text, or the conclusions are exceedingly short.
- W2. More importantly, there are not enough results to reproduce the experiments. There are no experimental details at all in the appendix that I could see, nor reference at how the training was performed or the hyperparameters selected (and, from what I've seen by eyeballing the attached code, the hyperparameters are identical for all experiments).
- W3. I am a bit concerned that all synthetic experiments follow exactly the same functional assumptions from the main paper. The authors should discuss whether assumptions like 3.3 can be met and consider violating those assumptions and/or run a sensitivity analysis.

I am happy to increase my rating once the authors clarify my concerns (especially W2).

---

> ### Author Rebuttal · Authors · 2026-03-31
>
> We sincerely thank the reviewer for the constructive feedback. We address the concerns below.
>
> >W1
>
> In Assumption 3.3, $\Psi_j: \mathbb{R} \rightarrow \mathbb{R}$ is an invertible and differentiable scalar function that wraps the affine combination $\beta_{j,1}(h_j) \cdot N_j + \beta_{j,0}(h_j)$. We will explicitly define this in the revised main text. Additionally, we will expand the Conclusion section to include a deeper discussion of the limitations identified during the review process.
>
> >W2
>
> Our implementation builds upon the Causal Normalizing Flow (CNF) architecture. Our base settings are: `layer_name`:Neural Autoregressive Flow, `max_epochs`:3000, `Batch_size`:1024, and `learning rate`:0.01 (with decay). We conducted a sensitivity analysis on 6 ER graphs (J=10).
>
> | Batch Size | Learning Rate | F1 Score (Detection) | F1 Score (Dissection) |
> | :--- | :--- | :--- | :--- |
> | 1024 | 0.01 | 1.000 | 1.000 |
> | 2048 | 0.01 | 1.000 | 1.000 |
> | 512 | 0.01 | 1.000 | 1.000 |
> | 1024 | 0.001 | 0.933 | 1.000 |
>
>
> Regarding the runner-up baseline PreDITEr, our initial experiments utilized `lambda_1`=0.05, `th_1`=0.35, `n_max_iter`=100. `lambda_1` controls the L1 sparsity penalty and `th_1` acts as a hard threshold for the estimated precision matrix difference. Our initial threshold (`th_1`=0.35) aimed to mitigate model misspecification. PreDITEr assumes linear Gaussian SEMs, but our non-linear shifts induce minor downstream covariance artifacts in the linear precision matrix difference. Given our large sample size (n=50,000), these artifacts can inflate the candidate set and cause false positives.
>
> We re-ran the PreDITEr baseline utilizing the hyperparameters in their original paper for large graph settings (`lambda_1`:0.1, `th_1`:1e-4, `n_max_iter`:500).
>
> (# of nodes=10,20,30,40,50)
> - ER: 0.940(0.115) | 0.978(0.050) | 0.954(0.077) |  0.965(0.057) | 0.963(0.040)
> - SF: 0.933(0.136) | 0.976(0.054) | 0.959(0.073) | 0.947(0.067) | 0.972(0.049)
>
> While this improved PreDITEr's performance, FANS maintains an edge on all SF graphs and on smaller ER graphs (J=10,20), with comparable performance on larger ER graphs.
>
> >W3: Same functional assumptions
>
> We evaluate our framework when Assumption 3.3 is violated. We introduced a squared noise term with a severity parameter $\lambda$.
>
> $X_j=\sum_k (\text{sin}([\text{PA}_j]_k^2))+\sum_k \sigma ([\text{PA}_j]_k^2)\cdot \varepsilon_j + \lambda (\sum_k [\text{PA}_j]_k^2)\cdot \varepsilon_j^2$ .
>
> With a mild violation ($\lambda=0.5$), the overall F1 score for dissecting function-only shifts from simultaneous shifts (following the same setting as Section 4.2) is 0.3748. Under a severe violation ($\lambda=1.0$), the F1 score is 0.3626.
>
> >Q1: Threshold Decision
>
> Our Appendix C contains the thresholding rules, dataset construction details, and runtime protocol.
>
> - Detection ($\tau$): According to Appendix C.1.1, for synthetic datasets, we employ an elbow point on the sorted Jensen-Shannon Divergence values.
>
> - Dissection: As detailed in Appendix C.1.1, we evaluate the distance correlation against a threshold of 0.05. In empirical settings, finite sample sizes yield a small non-zero correlation even under true independence. To account for this, the independence condition is considered violated when the correlation in Environment 2 exceeds the baseline empirical noise by this 0.05 threshold. Also, as suggested in 3.2, permutation tests based on distance covariances can be directly applied as an alternative for a more rigorous assessment of independence.
>
> >Q2
>
> Our method assumes the DAG of the base environment is known.
>
> We swapped the environments on 15 datasets (10-node ER graphs) where DAG remained identical. The swapped model maintained a `1.00` F1 score in both detection and dissection task. (Unswapped FANS also shows `1.00` F1 score on these 15 datasets.)
>
> >Q3
>
> The $T_{g_j}$ set can contain parent-dependent shifts: in the example $g_j(h, \varepsilon) = h \cdot \varepsilon$, setting $\delta(\text{pa}) = \text{pa}$ and $\psi(\varepsilon) = 2\varepsilon$ satisfies the condition in line 174 with non-constant $\delta$. The purpose of defining $T_{g_j}$ is to ensure a consistent dissection convention. Without it, the same observed change could be labeled as both a function shift and a noise shift simultaneously.
>
> >Q4
>
> If we train a second flow $\hat{T}^{(2)}$ on Environment 2, its inverse mapping is optimized to map a standard prior $\mathcal{N}(0,1)$ to $P(X^{(2)})$. Since the inferred noise from env1, $N^{(1)}=\hat{T}^{(1)}(X^{(1)})$, is forced to follow $\mathcal{N}(0,1)$, plugging $N^{(1)}$ into the second model simply feeds standard noise into a model designed to output $P(X^{(2)})$.
>
> >Q5 & Q6
>
> Algorithm 1 and Figure 2 are designed to provide a high-level overview of the FANS framework, so simultaneous shifts were omitted for clarity. Table 4 measures the total wall-clock time required for the Shift Detection task, while Table 5 measures the time for the Shift Dissection task.

---

> > ### Author Rebuttal · Reviewer_sEr5 · 2026-04-02
> >
> > Most of my concerns regard missing details and the writing, and after the rebuttal the authors have acknowledged these deficiencies and I trust they will work on them for the next revision.
> >
> > The minor methodological constraints I had were satisfactorily addressed.

---

> > > ### Author Response · Authors · 2026-04-02
> > >
> > > We sincerely thank you for the encouraging follow-up and for confirming that our responses have addressed your concerns. We will make sure all promised revisions are carefully incorporated into the camera-ready version.
> > >
> > > If you feel that the current responses and planned revisions sufficiently address the issues raised, we would be grateful if you could consider revisiting the overall score. We of course fully respect whatever decision you make. Thank you again for your time and valuable feedback.

---

### Decision · Program_Chairs · 2026-04-30

**Decision:**

Accept (regular)

**Comment:**

This paper proposes FANS, a framework to detect and dissect shifts in causal mechanisms across environments by training a causal normalizing flow on a base environment and using statistical tests to distinguish function shifts from noise shifts. The idea is considered novel and refreshing (sEr5), technically sound with reasonable assumptions (sEr5, zwKH, U6rR), and computationally appealing (oGUv). Although some weaknesses remain, including a lack of experimental details and a limited discussion of threshold calibration, fortunately, the authors have addressed the main issues during the rebuttal by clarifying experimental details, hyperparameters, and robustness considerations (sEr5, zwKH, U6rR).